# EVALUATING DATA INFLUENCE IN META LEARNING

**Chenyang Ren**[1,3,4,*]**, Huanyi Xie**[1,2,*]**, Shu Yang**[1,2]**, Meng Ding**[5]**, Dongrui Liu**[6]**,
**Lijie Hu**[3,†]**, Di Wang**[1,2,†]
[1]King Abdullah University of Science and Technology (KAUST)
[2]Provable Responsible AI and Data Analytics (PRADA) Lab
[3]Mohamed bin Zayed University of Artificial Intelligence (MBZUAI)
[4]Shanghai Jiao Tong University (SJTU)
[5]State University of New York at Buffalo    [6]Shanghai Artificial Intelligence Laboratory

## ABSTRACT

As one of the most fundamental models, meta learning aims to effectively address few-shot learning challenges. However, it still faces significant issues related to the training data, such as training inefficiencies due to numerous low-contribution tasks in large datasets and substantial noise from incorrect labels. Thus, training data attribution methods are needed for meta learning. However, the dual-layer structure of meta learning complicates the modeling of training data contributions because of the interdependent influence between meta parameters and task-specific parameters, making existing data influence evaluation tools inapplicable or inaccurate. To address these challenges, based on the influence function, we propose a general data attribution evaluation framework for meta learning within the bilevel optimization framework. Our approach introduces task influence functions (task-IF) and instance influence functions (instance-IF) to accurately assess the impact of specific tasks and individual data points in closed forms. This framework comprehensively models data contributions across both the inner and outer training processes, capturing the direct effects of data points on meta parameters as well as their indirect influence through task-specific parameters. We also provide several strategies to enhance computational efficiency and scalability. Experimental results demonstrate the framework's effectiveness in training data evaluation via several downstream tasks.

## 1 INTRODUCTION

Bilevel Optimization (BLO) has received significant attention and has become an influential framework in various machine learning applications including hyperparameter optimization (Sun et al., 2022; Okuno et al., 2021), data selection (Borsos et al., 2020; 2024), meta learning (Finn et al., 2017), and reinforcement learning (Stadie et al., 2020). A general BLO framework consists of two hierarchical optimization levels (outer and inner levels) and can be formulated as the following:

$$\min_{\lambda} f(\lambda, \theta^*(\lambda)) \quad s.t. \quad \theta^*(\lambda) \in \arg\min_{\theta'} g(\lambda, \theta'),$$

where the objective and variables of the outer-level problem $f$ are influenced by the inner-level problem $g$.

Among the various instantiations of BLO, meta learning has gained considerable interest due to its effectiveness in addressing the challenges of few-shot learning (Finn et al., 2017; Jamal & Qi, 2019; Hospedales et al., 2021; Franceschi et al., 2018; Yang et al., 2021). Meta learning involves two interdependent sets of parameters: meta parameters and task-specific parameters. Its inner level focuses on independently training across multiple few-shot tasks using the meta parameters $\lambda$ supplied by the outer level to derive the task-specific parameters $\theta^*(\lambda)$. The outer level, in turn, evaluates the performance of the inner level's task-specific parameters trained based on the meta

---

[*]The first two authors contributed equally to this work.
[†]Correspondence to Lijie Hu (Email Address: `lijie.hu@mbzuai.ac.ae`) and Di Wang (Email Address: `di.wang@kaust.edu.sa`).

parameter. Specifically, the outer level guides and assesses the training process of the inner level, while the outcomes of the inner level training provide task-specific model support for the outer level.

Recent advancements highlight the applicability of meta learning across various domains, including federated learning (Fallah et al., 2020) and multi-task learning (Wang et al., 2021), underscoring its versatility and potential. However, due to the large number of tasks, the training efficiency of meta learning often suffers, as these tasks do not contribute equally or effectively within extensive meta-datasets. This inefficiency highlights the need to discern the contributions of individual tasks to overall model training. Although various task selection methods have been proposed to improve training efficiency (Jamal & Qi, 2019; Liu et al., 2020; Achille et al., 2019), the dual-layer structure complicates the relationship between tasks and meta parameters, making analysis challenging. Additionally, the large datasets used in meta learning often contain some degree of noise, which poses a significant issue because model training typically assumes that all labels are accurate, and thus the presence of noisy data can severely hinder model performance (Ding et al., 2024). Furthermore, the two-layer bilevel structure complicates the understanding of learning processes and the interpretation of meta parameter predictions for specific tasks, undermining transparency and confidence—especially in high-stakes sectors like healthcare and finance (Kosan et al., 2023). Thus, it is necessary to develop methods for assessing data quality and attribution of meta learning.

Influence functions have emerged as a powerful tool for evaluating training data attribution as they efficiently estimate the contributions of data points using gradient information without requiring retraining, in contrast to the Shapley Value. However, it is essential to note that influence functions were initially designed for M-estimators (Huber, 1981), which limits their direct applicability to bilevel structures. Due to the bilevel structure, data in meta learning can be evaluated at two distinct levels: (1) the task level, which includes the training and validation sets for each task used to learn task-specific parameters and assess their effectiveness, and (2) the instance level, which focuses on the specific data points within the train or validation dataset of individual tasks. Each of these levels presents unique challenges for meta learning. First, unlike classical supervised learning models whose objective is a single-level optimization problem that is directly influenced by each training data point, the data in meta learning from each task influences the meta parameters directly and indirectly via task-specific parameters. While the influence function approach can be adapted to the meta-parameter by neglecting the inner-level training and indirect influences (as discussed in Section 4.1), this adaptation presents a challenge in accurately quantifying the data's attribution. Secondly, evaluating data is challenging because they do not explicitly appear in the training process of the outer meta parameter; instead, their influence is mediated through task-specific parameters.

To bridge the gap, in this paper, we propose an efficient data evaluation framework for meta learning (under the BLO setting). Specifically, we provide two methods: task-IF for measuring the influence of specific tasks in the meta learning dataset and instance-IF for measuring the influence of specific instances. For the task-level evaluation, our task-IF method emphasizes that the optimal conditions for model training are not only determined by the meta parameters but also incorporate the gradient information from task-specific parameters. At the instance level, we separately analyze data points in the validation and training datasets. We use a method similar to the task-IF to estimate data point impact for validation data. For training, we apply a novel two-stage closed-form estimation process across the model's inner and outer levels. First, we estimate the influence of instances on task-specific parameters, then introduce this influence at the outer level to approximate the effect of removing the instance. Our contribution can be summarized as follows:

- We conceptualize the data valuation problem in meta learning under the BLO framework, categorizing it into task and instance levels. At the instance level, we propose evaluation methods for training and validation samples. Specifically, we introduce a novel task-IF, which allows for precise modeling of task influence in meta learning by effectively capturing influences on both meta parameters and task-specific parameters. Additionally, we address the challenging instance-level evaluation through a two-stage estimation process that comprehensively accounts for both the direct and indirect effects of instances on meta parameters.

- To address the computational intensity and instability of inverse Hessian vector products, which have appeared several times in our method, we introduce several acceleration techniques, such as the EK-FAC method and Neumann expansion, to enhance computational efficiency and numerical stability, which allows for scaling the method to large-scale neural networks.

- Our method has potential applications in several real-world scenarios, including the automatic identification and removal of harmful tasks, and model editing, and the improvement of the interpretability of meta parameters. We verify the effectiveness and efficiency of task-IF and instance-IF through comprehensive experimental evaluation. Experimental results indicate that our framework performs well in data evaluation and editing of meta parameters.

## 2 RELATED WORK

**Meta learning.** Meta learning, often described as "learning to learn," has emerged as a pivotal approach for enabling models to rapidly adapt to new tasks by leveraging prior experience (Thrun & Pratt, 1998). (Finn et al., 2017) introduced Model-Agnostic Meta-Learning (MAML), framing the meta learning process as a bilevel optimization problem where the inner loop adapts model parameters to specific tasks, and the outer loop optimizes for performance across tasks. This framework has inspired a range of optimization-based meta learning methods (Finn & Levine, 2017; Finn et al., 2018; Li et al., 2017; Lee & Choi, 2018; Grant et al., 2018; Flennerhag et al., 2019). For example, Li et al. (2017) extended MAML to Meta-SGD, which learns not only the initial parameters but also the learning rates, enhancing adaptability across tasks. Grant et al. (2018) reformulate MAML as a method for probabilistic inference in a hierarchical Bayesian model. Finn et al. (2018) proposed Probabilistic MAML (PMAML), which incorporates uncertainty into the meta learning process by modeling the task distribution probabilistically. Flennerhag et al. (2019) introduced the Warped Gradient Descent, addressing the non-convergence due to bias and limited scale. Moreover, metric-based methods such as Matching Networks (Vinyals et al., 2016) and Prototypical Networks (Snell et al., 2017) focus on learning similarity metrics to classify new tasks effectively. Oreshkin et al. (2018b) advanced Prototypical Networks by proposing a task-specific adaptive scaling mechanism to learn a task-dependent metric.

**Influence function.** The influence function, initially a staple in robust statistics (Cook, 2000; Cook & Weisberg, 1980), has seen extensive adoption within machine learning since Koh & Liang (2017) introduced it to the field. Its versatility spans various applications, including detecting mislabeled data, interpreting models, addressing model bias, and facilitating machine unlearning tasks. Notable works in machine unlearning encompass unlearning features and labels (Warnecke et al., 2023), minimax unlearning (Liu et al., 2024), forgetting a subset of image data for training deep neural networks (Golatkar et al., 2020; 2021), graph unlearning involving nodes, edges, and features. Recent advancements, such as the LiSSA method (Agarwal et al., 2017; Kwon et al., 2023) and kNN-based techniques (Guo et al., 2021), have been proposed to enhance computational efficiency. Besides, various studies have applied influence functions to interpret models across different domains, including natural language processing (Han et al., 2020) and image classification (Basu et al., 2021), while also addressing biases in classification models (Wang et al., 2019), word embeddings (Brunet et al., 2019), and finetuned models (Chen et al., 2020).

Despite numerous studies on influence functions, we are the first to introduce this concept into bilevel optimization. In traditional models, the application of influence functions is relatively straightforward; however, in the bilevel optimization framework, it becomes significantly more complex due to the interactions between the variables across the outer and inner layers. This complexity necessitates a simultaneous consideration of the relationships between the outer objective function and the inner constraints. This paper provides a more comprehensive perspective on quantifying data influence and introduces, for the first time, a stratified approach to quantifying data within bilevel optimization.

## 3 PRELIMINARIES

**Notation.** For a twice differentiable function $L(\lambda, \theta(\lambda); D)$, $\partial_\lambda L(\lambda, \theta(\lambda); D)$ denotes the direct gradient (partial derivative) of $L$ w.r.t. $\lambda$ and $\partial_\theta L(\lambda, \theta(\lambda); D)$ denotes the direct gradient of $L$ w.r.t. $\theta(\lambda)$. And the total gradient (total derivative) of $L(\lambda, \theta(\lambda); D)$ w.r.t. $\lambda$ is calculated as $D_\lambda L(\lambda, \theta(\lambda); D) = \partial_\lambda L(\lambda, \theta(\lambda); D) + \frac{d\theta(\lambda)}{d\lambda} \cdot \partial_\theta L(\lambda, \theta(\lambda); D)$.

**Influence function.** The influence function (Huber, 1981) quantifies how an estimator relies on the value of each individual point in the sample. Consider a neural network $\hat{\theta} = \arg\min_\theta L(\theta, D) =$

$\arg\min_\theta \sum_{i=1}^n \ell(z_i; \theta)$ with loss function $\ell$ and dataset $D = \{z_i\}_{i=1}^n$. When an individual data point $z_m$ is removed from the training set, the retrained optimal retrained model is denoted as $\hat{\theta}_{-z_m}$. The influence function method provides an efficient way to approximate $\hat{\theta}_{-z_m}$ without the need of retraining. By up-weighing $z_m$-related term in the loss function by $\epsilon$, a series of $\epsilon$-parameterized optimal models $\hat{\theta}_{-z_m,\epsilon}$ will be obtained by $\hat{\theta}_{-z_m,\epsilon} = \arg\min_\theta [L(\theta, D) + \epsilon \cdot \ell(z_m; \theta)]$. Consider the term

$$\nabla L(\hat{\theta}_{-z_m,\epsilon}, D) + \epsilon \cdot \nabla \ell(z_m; \hat{\theta}_{-z_m,\epsilon}) = 0, \tag{1}$$

we perform a Taylor expansion at $\hat{\theta}$ and incorporate the optimal gradient condition at $\hat{\theta}_{-z_m}$ and $\hat{\theta}$: $\sum_{i=1}^n \nabla \ell(z_i; \hat{\theta}) + \epsilon \cdot \nabla \ell(z_m; \hat{\theta}) + H_{\hat{\theta}} \cdot \left( \hat{\theta}_{-z_m,\epsilon} - \hat{\theta} \right) \approx 0$, where $H_{\hat{\theta}} = \sum_{i=1}^n \nabla_{\hat{\theta}}^2 \ell(z_i; \hat{\theta})$ is the Hessian matrix. Consequently, the Influence Function is defined as the derivative of the change in parameters of the retrained model due to perturbation with respect to the perturbation: $\text{IF}(z_m) = \frac{d\hat{\theta}_{-z_m,\epsilon} - \hat{\theta}}{d\epsilon}\Big|_{\epsilon=0} \approx -H_{\hat{\theta}}^{-1} \cdot \nabla \ell(z_m; \hat{\theta})$. By setting $\epsilon = -1$, $z_m$ is completely removed from the retraining process. Then, $\hat{\theta}_{-z_m}$ can be approximated by a linear approximation as $\hat{\theta} - \text{IF}(z_m)$. Additionally, for a differentiable evaluation function, such as one used to calculate the total model loss over a test set, the change resulting from up-weighting $\epsilon$ to $z_m$ in the evaluation results can be approximated as $-\nabla f(\hat{\theta}) \cdot \text{IF}(z_m)$.

**Meta learning.** In this work, we consider the meta learning framework formulated as a bilevel optimization problem. Note that the BLO framework encapsulates most of the current objective functions in meta learning, such as MAML (Finn & Levine, 2017) and its variants (Finn et al., 2018; Nichol, 2018; Rusu et al., 2018; Killamsetty et al., 2022).

In meta-learning, the inner and outer objectives are typically defined by averaging the training and validation errors across multiple tasks. Given a set of tasks $\mathcal{I}$, we use a meta-training set $\mathcal{D} = \{D_i = D_i^{tr} \cup D_i^{val}\}_{i \in \mathcal{I}}$, where $D_i^{tr}$ and $D_i^{val}$ are the training and validation sets for each task $i$. The task-specific model parameter $\theta_i(\lambda)$ depends on the meta-parameter $\lambda$ and the training data, and is obtained by minimizing the inner objective: $\theta_i(\lambda) = \arg\min_{\theta_i} L_I(\lambda, \theta_i; D_i^{tr})$, where $L_I(\lambda, \theta_i; D_i^{tr})$ is the regularized empirical loss on $D_i^{tr}$ (with $\ell_2$-regularization to prevent overfitting). $L_I(\lambda, \theta_i; \mathcal{D}_i^{tr}) = \sum_{z \in \mathcal{D}_i^{tr}} \ell(z; \theta_i) + \frac{\delta}{2}\|\theta_i - \lambda\|^2$. And the $i$-th task related outer loss function is the empirical risk of $\lambda$ and $\theta_i$ on validation dataset $D_i^{val}$, defined as $L_O(\lambda, \theta_i(\lambda); \mathcal{D}_i^{val}) = \sum_{(x,y) \in \mathcal{D}_i^{val}} \ell(z; \theta_i(\lambda))$. Noting $\lambda$ here is shared between different tasks, and the outer objective is determined by calculating the average validation error on the parameters we obtained from the inner objective across multiple tasks as (we use $\ell_2$-regularization to avoid overfitting): $\lambda^* = \arg\min_\lambda L_{\text{Total}}(\lambda; \theta(\lambda); \mathcal{D}) = \arg\min_\lambda \sum_{i \in \mathcal{I}} L_O(\lambda, \theta_i(\lambda); D_i^{val}) + \frac{\delta}{2}\|\lambda\|^2$.

## 4 MODELING DATA INFLUENCE IN META LEARNING

It is evident that the datasets utilized for meta learning exhibit a hierarchical structure. Specifically, the whole dataset consists of multiple task-specific datasets. Consequently, when evaluating data influence, we can assess the influence of individual tasks on training outcomes while also conducting a detailed analysis of the impact of specific data points—i.e., instances—within a given task.

Due to the bilevel structure in meta-learning, it is essential to distinguish whether instances belong to the training or validation set. Consider the $k$-th task, if the data point we are concerned about is in the validation dataset, then only the meta parameter will be impacted, and we can directly apply the influence function method to evaluate its contribution. The data from task validation set directly influences the update of meta parameters. In contrast, the impact of the task training data on meta parameters is more complex; it indirectly affects the update of meta parameters by influencing the few-shot learning parameters that are relevant to the task. The following sections will provide our general frameworks for task and instance levels' data influence evaluation.

### 4.1 EVALUATING THE INFLUENCE OF TASKS

Consider the $k$-th task, the information related to this task includes $D_k = D_k^{tr} \cup D_k^{val}$. We will leverage influence function to evaluate how $D_k$ will affect the model given by meta learning. When

we remove the $k$-th task from meta learning, the $k$-th inner training task will be completely eliminated. This is reflected in the outer layer, where $L_O\left(\lambda, \theta_k(\lambda); D_k^{val}\right)$ will disappear. Therefore, we can ignore the specifics of the $k$-th inner training and focus solely on the outer level. The retrained meta parameter is defined as $\lambda^*_{-k} = \arg\min_\lambda \sum_{i\neq k} L_O\left(\lambda, \theta_i(\lambda); D_i^{val}\right) + \frac{\delta}{2}\|\lambda\|^2$.

**A direct approach.** If we directly adopt the same idea of the original influence function and consider $D_k$ as a data point, we then may up-weight the loss for the $k$-th task $L_O\left(\lambda, \theta_k(\lambda); D_k^{val}\right)$ by a small perturbation $\epsilon$ and replace the $\lambda^*_{-k}$ by the minimizer of the $\epsilon$-parameterized model $\lambda^*_\epsilon$. Ignoring the dependence between the task-specific parameters and the meta parameters, one can intuitively derive an influence function by using a similar Taylor expansion of the partial derivative w.r.t. $\lambda$ of the outer objective function as

$$\text{IF}(D_k) = -\left(\delta \cdot I + H_{\lambda,\text{direct}}\right)^{-1} \cdot \partial_\lambda L_O\left(\lambda^*, \theta_k(\lambda^*); D_k^{val}\right), \tag{2}$$

where $H_{\lambda,\text{direct}}$ is the Hessian matrix directly with respect to $\lambda$, defined as

$$H_{\lambda,\text{direct}} = \sum_{i\in\mathcal{I}} \partial_\lambda\partial_\lambda L_O\left(\lambda^*, \theta_i(\lambda^*); D_i^{val}\right). \tag{3}$$

**Our method.** However, this method is flawed as it overlooks an important aspect: $\theta_i(\lambda)$ is dependent on $\lambda$, and the experimental results in Table 1 further demonstrate the inaccuracy of this method. Mathematically, in the original IF definition, since there is no inner level function, we have $\nabla L(\hat{\theta}; D) = 0$ and (1). However, as in bilevel optimization the outer objective function $L_{\text{Total}}$ dependents both $\lambda$ and $\{\theta_i(\lambda)\}_{i\in\mathcal{I}}$, for $\lambda^*$ we do not have the direct gradient $\partial_\lambda L_{\text{Total}}(\lambda^*, \theta(\lambda^*); \mathcal{D}) = 0$ and its corresponding (1). Thus, we cannot use a similar Taylor expansion to (1) and get (2).

To address the above issue, the key observation is that while the direct gradient at $\lambda^*$ is non-zero, its corresponding total gradient of the loss function is zero, i.e., $D_\lambda L_{\text{Total}}(\lambda^*, \theta(\lambda^*); \mathcal{D}) = 0$. The following result provides a result on the computation of the total gradient.

**Theorem 4.1.** *The total gradient of the $i$-th task related outer loss $L_O\left(\lambda, \theta_i(\lambda); D_i^{val}\right)$ with respect to $\lambda$ can be written as:* $D_\lambda L_O\left(\lambda, \theta_i(\lambda); D_i^{val}\right) = \partial_\lambda L_O\left(\lambda, \theta_i(\lambda); D_i^{val}\right) + \frac{d\theta_i(\lambda)}{d\lambda} \cdot \partial_{\theta_i} L_O\left(\lambda, \theta_i(\lambda); D_i^{val}\right)$. *The term $\frac{d\theta_i(\lambda)}{d\lambda}$ can be calculated by*

$$\frac{d\theta_i(\lambda)}{d\lambda} = -\partial_\lambda\partial_{\theta_i} L_I\left(\lambda, \theta_i(\lambda); \mathcal{D}_i^{tr}\right) \cdot H_{i,in}^{-1}, \tag{4}$$

*where $H_{i,in}$ is the $i$-th inner-level Hessian matrix, defined as $H_{i,in} = \partial_{\theta_i}\partial_{\theta_i} L_I\left(\lambda, \theta_i(\lambda); \mathcal{D}_i^{tr}\right)$.*

Based on the discussion above, it is clear that the direct Hessian matrix in (3) is also inadequate for the influence function in the bilevel setting. Therefore, we propose an expression for the total Hessian matrix that incorporates the total derivative.

**Definition 4.2.** The total Hessian matrix of the outer total loss $L_{\text{Total}}(\lambda, \theta(\lambda); \mathcal{D})$ with respect to $\lambda$ is defined as $H_{\lambda,\text{Total}} = \sum_{i\in\mathcal{I}}\left(D_\lambda D_\lambda L_O\left(\lambda, \theta_i(\lambda); D_i^{val}\right)\right)$.

Based on the total gradient and the total Hessian matrix we proposed, we can up weight the $L_O(\lambda, \theta_k; \mathcal{D}_k^{val})$ term at the outer level and derive its influence function.

**Theorem 4.3.** *(task-IF) Define the task-IF of the $k$-th task in the meta parameter $\lambda^*$ as*

$$\text{task-IF}(D_k; \lambda^*, \theta_k(\lambda^*)) = -\left(\delta \cdot I + H_{\lambda,Total}\right)^{-1} \cdot D_\lambda L_O\left(\lambda^*, \theta_k(\lambda^*); D_k^{val}\right).$$

*Then after the removal of the $k$-th task, the retrained meta parameter $\lambda^*_{-k}$ can be estimated by $\lambda^*_{-k} \approx \lambda^* - \text{task-IF}(D_k; \lambda^*, \theta_k(\lambda^*))$.*

*Remark* 4.4. Note that the task influence function depends on the total Hessian matrix. For simplicity, we denote $L_O\left(\lambda, \theta_i(\lambda); D_i^{val}\right)$ as $L_O^i$, then it can be written as

$$H_{\lambda,\text{Total}} = \sum_{i\in\mathcal{I}}\left(\partial_\lambda\partial_\lambda L_O^i + 2\frac{d\theta_i(\lambda)}{d\lambda}\partial_{\theta_i}\partial_\lambda L_O^i + \frac{d^2\theta_i(\lambda)}{d\lambda^2}\partial_{\theta_i} L_O^i + \left(\frac{d\theta_i(\lambda)}{d\lambda}\right)^2 \partial_{\theta_i}\partial_{\theta_i} L_O^i\right).$$

Due to the complex definition of $\frac{d\theta_i(\lambda)}{d\lambda}$ as discussed in (4), $\frac{d^2\theta_i(\lambda)}{d\lambda^2}$ requires a three-order partial derivative, presenting significant computation challenges. To address these complexities, we will introduce several accelerated algorithms in Section 5.1.

### 4.2 Evaluating the influence of individual instance

There are two instances in the meta learning dataset: instances in the training dataset and instances in the validation dataset. The validation instance appears on the outer level and will only influence the meta parameter without directly impacting the task-specific parameter. The training instances appear at the inner level and impact the task-specific parameter, thereby influencing the meta parameter indirectly. Therefore, we must delve into the inner training process and conduct a two-stage evaluation. In a word, the evaluation of validation instances is similar to the task evaluation problem, while evaluating training instances is more challenging and requires new techniques.

**Validation data influence.** We first focus on the evaluation of influence for the validation data $\tilde{z} = (\tilde{x}, \tilde{y})$ of the $k$-th task. Since the validation data was not used in the inner level loss, we can directly follow the idea of task-IF to the outer loss function. The only difference is that we up-weigh a term related to data point $\tilde{z}$ in the $k$-th task $\ell(\tilde{z}; \theta_k(\lambda))$ rather than the task term $L_O(\lambda, \theta_k(\lambda); \mathcal{D}_k^{val})$. Then similar to Theorem 4.3, we obtain the following theorem.

**Theorem 4.5** (Instance-IF for Validation Data). *The influence function of the validation data* $\tilde{z} = (\tilde{x}, \tilde{y})$ *is instance-IF*$(\tilde{z}; \lambda^*, \theta_k(\lambda^*)) = -\left(\delta \cdot I + H_{\lambda, Total}\right)^{-1} \nabla \ell(\tilde{z}; \theta_k(\lambda^*))$. *After the removal of* $\tilde{z}$, *the retrained optimal meta parameter* $\lambda^*_{-\tilde{z}}$ *can be estimated by* $\lambda^*_{-\tilde{z}} \approx \lambda^* -$ *instance-IF*$(\tilde{z}; \lambda^*, \theta_k(\lambda^*))$.

**Training data influence.** We evaluate the influence of training data first. We will employ a two-phase analysis to analyze the impact of an individual data point $\tilde{z} = (\tilde{x}, \tilde{y})$ in the $k$-th task. Initially, we utilize the IF to assess the data point's effect on its corresponding task-related parameter $\theta_k(\lambda)$. Since the meta parameter, $\lambda$ is passed from the outer level and remains fixed during each inner-level training period. Thus, we can use the classical influence function directly to $L_I(\lambda, \theta_k; \mathcal{D}_k^{tr})$.

**Theorem 4.6.** *The influence function for the inner level (inner-IF) of* $\tilde{z} = (\tilde{x}, \tilde{y})$ *from the training set is inner-IF*$(\tilde{z}; \theta_k(\lambda^*)) = -H_{k,in}^{-1} \cdot \nabla_{\theta_k} \ell(\tilde{z}; \theta_k(\lambda^*))$, *where* $H_{k,in} = \partial_{\theta_k} \partial_{\theta_k} L_I(\lambda^*, \theta_k(\lambda^*); \mathcal{D}_k^{tr})$ *is the $k$-th inner-level Hessian matrix. After the removal of* $\tilde{z}$, *denote* $\theta_k^-(\lambda^*)$ *as its corresponding retrained task-parameter* $\theta_k^-(\lambda^*) = \arg\min_{\theta_i} L_I(\lambda^*, \theta_i; D_i^{tr}\backslash\tilde{z})$. *Then it can be estimated by* $\theta_k^-(\lambda^*) \approx \theta_k(\lambda^*) -$ *inner-IF*$(\tilde{z}; \theta_k(\lambda^*))$.

Now we aim to approximate the influence of replacing $\theta_k$ by $\theta_k^-$ in the outer level. That is, how will the optimal meta parameter $\lambda^*$ change under the removal of $\tilde{z}$. Note that the outer level loss only depends on the validation data and remains unchanged after the removal of $\tilde{z}$. This challenges applying up-weighting to the influence function effectively as the change of the outer loss is implicit. The result, which is shown in Theorem 4.7, shows the change in the outer-level loss function resulting from the removal of $\tilde{z}$.

**Theorem 4.7.** *If the $k$-th task related parameter changes from $\theta_k$ to $\theta_k^-$, we can estimate the related loss function difference* $L_O\left(\left(\lambda, \theta_k^-(\lambda)\right); D_k^{val}\right) - L_O\left(\lambda, \theta_k(\lambda); D_k^{val}\right)$ *by* $P(\lambda, \theta_k(\lambda); \tilde{z})$, *which is defined as* $-\nabla_\theta L_O\left(\lambda, \theta_k(\lambda); D_k^{val}\right)^{\mathrm{T}} \cdot$ *inner-IF*$(\tilde{z}; \theta_k(\lambda))$.

Now we can derive the influence function. Motivated by the above result, instead of up-weighting the loss function term, we focus on up-weighting the data influence term $P(\lambda, \theta_k(\lambda); \tilde{z})$. We add this term into the outer level loss function and up-weight it by a small $\zeta$, which varies from 0 to 1. Then the outer level loss function becomes $\lambda^*_{-\tilde{z},\zeta} = \arg\min \sum_{i \in \mathcal{I}} L_{Total}(\lambda; \theta(\lambda); \mathcal{D}) + \zeta \cdot P(\lambda, \theta_k(\lambda); \tilde{z})$. When $\zeta = 1$, the influence of $\tilde{z}$ is fully removed by adding this term into the loss function.

**Proposition 4.8** (Instance-IF for Training Data). *Let $\zeta \to 0$, we can obtain the influence function of the outer level as instance-IF*$(\tilde{z}; \lambda^*, \theta_k(\lambda^*)) = -H_{\lambda, Total}^{-1} \cdot \mathrm{D}_\lambda P(\lambda^*, \theta_k(\lambda^*); \tilde{z})$. *After the removal of $\tilde{z}$, the optimal meta parameter $\lambda^*_{-\tilde{z}}$ can be estimated by* $\lambda^*_{-\tilde{z}} \approx \lambda^* -$ *instance-IF*$(\tilde{z}; \lambda^*, \theta_k(\lambda^*))$.

## 5 Computation acceleration and practical applications

### 5.1 Computation acceleration

Although the results in Section 4 are closed-form solutions for evaluating data influence, they involve extensive calculations of the inverse Hessian-vector product (iHVP), which are coupled with other

Table 1: Performance comparison of different methods.

| Evaluation Level | Method | omniglot | | MNIST | | MINI-Imagenet | | FC100 | |
|---|---|---|---|---|---|---|---|---|---|
| | | Accuracy | RT (second) | Accuracy | RT (second) | Accuracy | RT (second) | Accuracy | RT (second) |
| Task | Retrain | 0.7908±0.0061 | 316.74±8.45 | 0.6373±0.0311 | 244.24±2.18 | 0.3071±0.0460 | 682.56±7.91 | 0.3599±0.0002 | 362.02±0.01 |
| | Task-IF(Task-IF) | 0.7804±0.0116 | 65.69±3.41 | 0.6118±0.0469 | 45.72±0.46 | 0.2668±0.0322 | 74.63±1.40 | 0.3403±0.0485 | 56.20±1.71 |
| | Direct-IF(Direct-IF) | 0.5422±0.0292 | 7.46±0.05 | 0.3247±0.0224 | 7.37±0.04 | 0.2214±0.0136 | 10.24±0.05 | 0.2906±0.0013 | 10.63±0.01 |
| | EKFAC IF | 0.4872±0.0001 | 631.90±10.05 | 0.4764±0.0123 | 630.96±20.02 | 0.1868±0.0106 | 1937.42±41.25 | 0.3133±0.0021 | 648.41±30.21 |
| | TRAK | 0.3076±0.0001 | 1022.52±3.49 | 0.4160±0.0001 | 1020.60±0.01 | 0.2592±0.0270 | 1154.17±0.17 | 0.3242±0.0004 | 2776.14±25.34 |
| | Tracin | 0.7690±0.0001 | 375.42±0.65 | 0.5844±0.0012 | 376.85±0.63 | 0.2592±0.0270 | 1154.16±0.1757 | 0.2607±0.0002 | 481.37±8.76 |
| Instance | Retrain(Training) | 0.7852±0.0080 | 251.76±1.21 | 0.6540±0.0148 | 252.80±1.07 | 0.3263±0.0574 | 392.24±0.40 | 0.3549±0.0058 | 361.9944±0.0078 |
| | Instance-IF(Training) | 0.7686±0.0037 | 4.49±0.02 | 0.6123±0.0225 | 4.53±0.02 | 0.2854±0.0424 | 7.17±0.03 | 0.3645±0.0001 | 7.2277±0.0829 |
| | Retrain(Validation) | 0.7895±0.0080 | 260.48±7.62 | 0.6265±0.0410 | 210.76±93.10 | 0.3222±0.0601 | 358.49±13.08 | 0.3651±0.0030 | 360.73±0.06 |
| | Instance-IF(Validation) | 0.7790±0.0106 | 5.53±0.06 | 0.5476±0.0924 | 5.71±0.51 | 0.2767±0.0350 | 7.47±0.83 | 0.3698±0.0023 | 7.2141±0.04 |

computational steps and also necessitate the computation of third-order partial derivatives. Thus, developing acceleration techniques is essential to enhance the scalability of our methods. We now present approximation methods designed to accelerate iHVP and third-order partial derivatives.

**iHVP acceleration via EK-FAC.** Due to the specific method for calculating the derivative of task-related parameter $\theta_i(\lambda)$ in (4) and the iHVP calculation in Theorem 4.6, our algorithm involves the computation of iHVP at several different stages. To expedite this computation and eliminate the need to store the Hessian matrix explicitly, we employ the EK-FAC method (Kwon et al., 2023). Additional implementation details can be found in the Appendix.

**Total Hessian matrix approximation.** As mentioned in Remark 4.4, computing the total Hessian matrix for large-scale neural networks presents significant challenges, primarily due to the requirement of calculating the third-order partial derivatives of the loss function with respect to the parameters. Given that the information conveyed by third-order derivatives is relatively limited compared to that of first and second-order derivatives, we propose an approximation for the total Hessian.

**Theorem 5.1.** *To approximate total Hessian $H_{\lambda,Total}$, we can replace the $H'$ term in total Hessian (see 4.4), defined as $H' = \frac{\mathrm{d}^2\theta_i(\lambda)}{\mathrm{d}\lambda^2} \cdot \partial_{\theta_i} L_O^i + \left(\frac{\mathrm{d}\theta_i(\lambda)}{\mathrm{d}\lambda}\right)^2 \cdot \partial_{\theta_i}\partial_{\theta_i} L_O^i$ by $\Gamma = \sum_{i\in\mathcal{I}} \|\partial_\theta L_O^i\|_1 \cdot I$, where $I$ is the identity matrix.*

Generally, compared to the original term $H'$, $\Gamma$ reflects the extent to which each parameter influences the loss function and captures significant gradient information by focusing on the main influencing factors via ignoring the second-order derivative terms. This enables the practical simplification of complex mathematical expressions while preserving essential information. However, EK-FAC is no longer applicable for computing the iHVP for the total Hessian proposed in this paper. This is because EK-FAC is based on the assumption that the parameters between different layers are independent. In contrast, the definition of the total Hessian in (4.4) needs interactions between different layers. Therefore, EK-FAC may be insufficiently accurate here. To compute the iHVP precisely, efficiently, and stably, we propose the following accelerated method:

**iHVP acceleration via EK-FAC.** Due to the specific method for calculating the derivative of task-related parameter $\theta_i(\lambda)$ in (4) and the iHVP calculation in Theorem 4.6, our algorithm involves the computation of iHVP at several different stages. To expedite this computation and eliminate the need to store the Hessian matrix explicitly, we employ the EK-FAC method (Kwon et al., 2023). Additional implementation details can be found in the Appendix.

**Neumann series approximation method.** For a vector $v$, we can express $\tilde{H}_{\lambda,\text{Total}}^{-1} \cdot v$ through the Neumann series: $\tilde{H}_{\lambda,\text{Total}}^{-1} \cdot v = v + \sum_{j=1}^{+\infty}(I - \tilde{H}_{\lambda,\text{Total}})^j \cdot v$. By truncating this series at order $J$, we derive the following approximation: $\tilde{H}_{\lambda,\text{Total}}^{-1} \cdot v \approx v + (I - \tilde{H}_{\lambda,\text{Total}}) \cdot v + \cdots (I - \tilde{H}_{\lambda,\text{Total}})^J \cdot v$. It is important to note that we do not specifically accelerate the inversion of the Hessian matrix; rather, we directly optimize the iHVP procedure. As a result, throughout the entire computation process, there is no need to simultaneously store the large Hessian matrix, which significantly reduces the memory requirements of our method.

## 5.2 APPLICATIONS OF INFLUENCE FUNCTIONS

In this section, we present application methods for downstream tasks based on the previously derived influence functions for tasks and instances.

**Model editing by data removal.** We can use IFs to update the model under the removal of certain data or tasks. Specifically, the model after removing the $k$-th task can be estimated as $\lambda^* - \text{task-IF}(D_k; \lambda^*, \theta_k(\lambda^*))$. Additionally, the model for removing instance $\tilde{z}$ from the $k$-th task

can be estimated as $\lambda^* - \text{instance-IF}(\tilde{z}; \lambda^*, \theta_k(\lambda^*))$.

**Data evaluation** By appropriately selecting the model evaluation function, we can define influence scores(ISs), which measure the influence of specific tasks or instances on new tasks. These scores can then be applied to task selection in order to enhance meta learning performance or to adversarial attacks to assess model robustness.

**Definition 5.2** (Evaluation function for meta learning). Given a new task specific dataset $D_{new} = D_{new}^{tr} \cup D_{new}^{val}$, the few-shot learned parameter $\theta'$ for the task is obtained by minimizing the loss $L_I(\lambda^*, \theta; D_{new}^{tr})$. We can evaluate the performance of the meta parameter $\lambda^*$ based on the loss of $\theta'$ on the validation dataset, expressed as $\sum_{z \in D_{new}^{val}} \ell(z; \theta')$.

Based on this metric, we propose a calculation method for the task and instance Influence Score, defined as $\text{IS} = \sum_{z \in D_{new}^{val}} \nabla_{\theta'} \ell(z; \theta') \text{IF}$. A detailed theoretical derivation can be found in Appendix.

## 6 EXPERIMENTS

In this section, we demonstrate our main experimental results on utility, efficiency, effectiveness, and the ability to identify harmful data.

### 6.1 EXPERIMENTAL SETTINGS

**Dataset and evaluation metric.** We utilize 4 datasets: *omniglot* (Lake et al., 2015), *MNIST* (LeCun et al., 1998), *MINI-Imagenet* (Vinyals et al., 2016) and *FC100* (Oreshkin et al., 2018a). We report two metrics: Accuracy and runtime (RT). *Accuracy* is the proportion of correct predictions, and *Runtime* (seconds) measures the time to update the model. See Appendix D for details.

**Baselines.** At the task level, besides the ground truth-retraining, we compare direct IF and Task-IF with three baseline data attribution evaluation methods: EKFAC IF (Grosse et al., 2023), TARK (Park et al., 2023), and TracIN (Pruthi et al., 2020). *Direct IF*, as described in Section 4.1, is a direct implementation of (2) and a generalization of the original IF. At the instance level, we use retraining as the ground truth. *Retrain (Training/Validation)* involves retraining the model after removing a selected instance from the training or validation dataset for a task-specific dataset. Additionally, we conduct experiments using instance-IF (Training/Validation). *Instance-IF (Training)* and *Instance-IF (Validation)* directly implement the theoretical results from Proposition 4.8 and Theorem 4.5, respectively. Detailed algorithms for all data evaluation methods are provided in the Appendix.

### 6.2 EVALUATION OF UTILITY AND EDITING EFFICIENCY

Task-IF and direct-IF can be utilized for retraining surrogates and data attribution. To assess their utility and editing efficiency, we compare our methods with other baselines from the perspectives of accuracy and runtime. Specifically, we employ various data evaluation techniques to identify and remove 40% of harmful data from the training dataset. Subsequently, the model is retrained to evaluate its test accuracy. Notably, task-IF and direct-IF can also be used to update the model directly, eliminating the need for retraining. Our results in Table 1 highlight the superiority of task-IF over baselines, while highlighting the critical trade-off between computational efficiency and accuracy. On the Omniglot dataset, task-IF achieved accuracy comparable to retraining (0.7804 vs. 0.7908) while reducing runtime from 316.74 to 65.69. Similarly, on MINI-ImageNet, task-IF cut runtime from 682.56 to 74.63 with a modest accuracy trade-off (0.3071 vs. 0.2668). These findings demonstrate task-IF's ability to save 75%-80% of runtime while maintaining competitive accuracy. While direct-IF achieves the fastest runtime among all approaches, it shows significant degradation in accuracy across all datasets. This is due to the fact that it ignores the dependence between the task-specific parameters and the meta parameters. As we mentioned in Theorem 4.1, our task-IF leverages the total gradient and Hessian to fill the gap, which will also slightly increase the computation time due to their addition terms.

For the instance-level evaluation, we observe that instance-IF demonstrates promising results, particularly in the scenario where we aim to evaluate the attribution of training samples. The approach maintains relatively stable performance between training and validation cases, with accuracy scores

of 0.7686 and 0.7790, respectively, on omniglot, while achieving remarkable runtime efficiency (4.49 and 5.53 seconds). This sight difference in model accuracy between IF method and retraining may be attributed to different factors. For instance, our theoretical framework relies on the stability of influence estimation across training, which may vary depending on the model's convergence behavior. Moreover, the effectiveness of instance-level influence computation might be affected by the inherent complexity and feature distribution of different datasets, explaining the varying performance gaps observed between MNIST and MINI-ImageNet.

### 6.3 RESULTS OF HARMFUL DATA REMOVAL

We also use influence functions to assess how helpful or harmful each task is. To validate this, we create harmful tasks by flipping labels in 80% of tasks and 40% of MNIST samples. Figure 1 reports test accuracy and the fraction of mislabeled tasks detected as we vary the proportion of training data checked. "IS" removes tasks by influence score; "Random" removes tasks at random. The left plot shows that our IS-based approach consistently

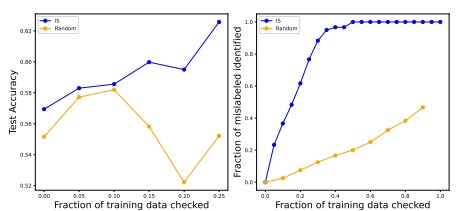

Figure 1: Harmful tasks removal experiment.

outperforms random removal across all checking ratios. When examining 25% of the training data, the IS-based method achieves a test accuracy of approximately 62.5%, while random removal yields only 55% accuracy. This significant performance gap of 7.5 percentage points indicates that our method effectively identifies and removes truly harmful tasks rather than arbitrary ones. The right plot further validates our approach's effectiveness in identifying mislabeled tasks. The IS-based method demonstrates remarkably efficient detection, identifying nearly 100% of mislabeled tasks after checking only 40% of the training data. In contrast, the random removal baseline shows a linear relationship between the fraction checked and identified, achieving only about 45% detection rate even after examining 95% of the training data. This substantial difference in detection efficiency (approximately $2.2\times$ better) underscores our method's strong capability in targeting harmful tasks.

### 6.4 EFFECTIVENESS OF IF

Here, we aim to show that our proposed influence functions can indeed be used to attribute data. To investigate the effectiveness, we first calculate the influence scores for all tasks or instances in the training set based on our proposed task-IF and instance-IF. Using these scores, we iteratively remove tasks or instances in descending order of influence score and subsequently retrain the model to examine its robustness. As shown in Figure 2, removing high-influence tasks or instances significantly reduces test accuracy. At the task level, accuracy drops from about 0.62 to 0.58, while at the instance level, it falls from roughly 0.40 to 0.25. In comparison, randomly removing data has a much smaller impact, highlighting the effectiveness of targeted removal based on influence scores. These results suggest that the influence score calculation effectively identifies critical data points, with high-influence tasks or instances playing a piv-

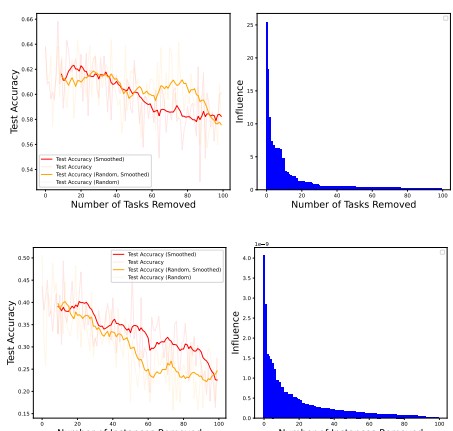

Figure 2: MNIST Data Attribution (up: Task, below: Instance).

otal role in maintaining model performance. This analysis demonstrates the effectiveness of influence scores in pinpointing valuable data. Additional visualizations are provided in Appendix E.

## 7 CONCLUSION

This paper presents data attribution evaluation methods for meta learning, addressing challenges like training inefficiencies and noise from mislabeled data. By introducing task-IF and instance-IF, we offer a comprehensive approach for evaluating data across tasks and instances. Our framework

captures both direct and indirect impacts of data points on meta parameters, improving efficiency and scalability.

## ACKNOWLEDGMENT

Di Wang, Huanyi Xie, Chenyang Ren, and Shu Yang are supported in part by the funding BAS/1/1689-01-01,RGC/3/7125-01-01, FCC/1/5940-20-05, FCC/1/5940-06-02, and King Abdullah University of Science and Technology (KAUST) – Center of Excellence for Generative AI, under award number 5940 and a gift from Google. Lijie Hu is supported by the funding BF0100 from Mohamed bin Zayed University of Artificial Intelligence (MBZUAI). For computer time, this research used lbex managed by the Supercomputing Core Laboratory at King Abdullah University of Science & Technology (KAUST) in Thuwal, Saudi Arabia.

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

## A    LLM USAGE

We used LLMs to refine grammar and improve language fluency. The authors reviewed and edited all LLM-generated content and assume full responsibility for the final text.

## B    OMITTED PROOFS

### B.1    EVALUATING THE INFLUENCE OF TASKS

**Theorem B.1.** *The total gradient of the $i$-th task related outer loss $L_O\left(\lambda, \theta_i(\lambda); D_i^{val}\right)$ with respect to $\lambda$ can be written as:*

$$
\begin{aligned}
&\mathrm{D}_\lambda L_O\left(\lambda, \theta_i(\lambda); D_i^{val}\right) \\
&=\partial_\lambda L_O\left(\lambda, \theta_i(\lambda); D_i^{val}\right) + \frac{\mathrm{d}\theta_i(\lambda)}{\mathrm{d}\lambda} \cdot \partial_{\theta_i} L_O\left(\lambda, \theta_i(\lambda); D_i^{val}\right).
\end{aligned}
$$

*The term $\frac{\mathrm{d}\theta_i(\lambda)}{\mathrm{d}\lambda}$ can be calculated by*

$$
\frac{\mathrm{d}\theta_i(\lambda)}{\mathrm{d}\lambda} = -\partial_\lambda \partial_{\theta_i} L_I\left(\lambda, \theta_i(\lambda); \mathcal{D}_i^{tr}\right) \cdot H_{i,in}^{-1}, \tag{5}
$$

*where $H_{i,in}$ is the $i$-th inner-level Hessian matrix, defined as $H_{i,in} = \partial_{\theta_i} \partial_{\theta_i} L_I\left(\lambda, \theta_i(\lambda); \mathcal{D}_i^{tr}\right)$.*

*Proof.* From the Chain Rule, the total gradient of $L_O\left(\lambda, \theta_i(\lambda); D_i^{val}\right)$, denoted as $\mathrm{D}_\lambda L_O\left(\lambda, \theta_i(\lambda); D_i^{val}\right)$, is calculated by:

$$
\begin{aligned}
&\mathrm{D}_\lambda L_O\left(\lambda, \theta_i(\lambda); D_i^{val}\right) \\
&=\partial_\lambda L_O\left(\lambda, \theta_i(\lambda); D_i^{val}\right) + \frac{\mathrm{d}\theta_i(\lambda)}{\mathrm{d}\lambda} \cdot \partial_{\theta_i} L_O\left(\lambda, \theta_i(\lambda); D_i^{val}\right).
\end{aligned}
$$

We will begin to derive the calculation method for $\frac{\mathrm{d}\theta_i(\lambda)}{\mathrm{d}\lambda}$ term. From the optimal condition of the $i$-th inner level: $\partial_\theta L_I(\lambda, \theta_i(\lambda); D_i^{val}) = 0$. Then take derivative respect to $\lambda$

$$
\partial_\lambda \partial_\theta L_I(\lambda, \theta_i(\lambda); D_i^{tr}) + \frac{\mathrm{d}\theta_i(\lambda)}{\mathrm{d}\lambda} \cdot \partial_\theta \partial_\theta L_I(\lambda, \theta_i(\lambda); D_i^{tr}) = 0
$$

Then from Implicit Function Theorem, we can calculate $\frac{\mathrm{d}\theta_i(\lambda)}{\mathrm{d}\lambda}$ term as:

$$
\frac{\mathrm{d}\theta_i(\lambda)}{\mathrm{d}\lambda} = -\partial_\lambda \partial_\theta L_I(\lambda, \theta_i(\lambda); D_i^{tr}) \cdot \partial_\theta \partial_\theta L_I(\lambda, \theta_i(\lambda); D_i^{tr})^{-1}
$$

Noting here $\partial_\theta \partial_\theta L_I(\lambda, \theta_i(\lambda); D_i^{tr})$ is the Hessian matrix of the $i$-th inner level loss w.r.t $\theta_i(\lambda)$, we denote it as $H_{i,in}$. Then, the final expression of $\frac{\mathrm{d}\theta_i(\lambda)}{\mathrm{d}\lambda}$ is:

$$
\frac{\mathrm{d}\theta_i(\lambda)}{\mathrm{d}\lambda} = -\partial_\lambda \partial_{\theta_i} L_I\left(\lambda, \theta_i(\lambda); D_i^{tr}\right) \cdot H_{i,in}^{-1}.
$$

$\square$

**Definition B.2.** The total Hessian matrix of the outer total loss $L_{\text{Total}}(\lambda, \theta(\lambda); \mathcal{D})$ with respect to $\lambda$ is defined as:

$$
H_{\lambda, \text{Total}} = \sum_{i \in \mathcal{I}} \left(\mathrm{D}_\lambda \mathrm{D}_\lambda L_O\left(\lambda, \theta_i(\lambda); D_i^{val}\right)\right).
$$

**Theorem B.3.** *(task-IF) Define the task-IF of the $k$-th task in the meta parameter $\lambda^*$ as*

$$
\begin{aligned}
&\text{task-IF}(D_k; \lambda^*, \theta_k(\lambda^*)) \\
&= -\left(\delta \cdot I + H_{\lambda, Total}\right)^{-1} \cdot \mathrm{D}_\lambda L_O\left(\lambda^*, \theta_k(\lambda^*); D_k^{val}\right).
\end{aligned}
$$

*Then after the removal of the $k$-th task, the retrained meta parameter $\lambda_{-k}^*$ can be estimated by*

$$
\lambda_{-k}^* \approx \lambda^* - \text{para-IF}(D_k; \lambda^*, \theta_k(\lambda^*)).
$$

*Proof.* The outer objective is determined by calculating the average validation error on the task-specific parameters we obtained from the inner objective across multiple tasks as:

$$\lambda^* = \arg\min_{\lambda} \sum_{i \in \mathcal{I}} L_O\left(\lambda, \theta_i(\lambda); D_i^{val}\right) + \frac{\delta}{2}\|\lambda\|^2. \tag{6}$$

We upweight the term in the total loss function (6) related to task $k$ by $\epsilon$ and obtain

$$\lambda_\epsilon \triangleq \arg\min_{\lambda} \sum_{i \in \mathcal{I}} L_O(\lambda, \theta_i; D_i^{val}) + \epsilon \cdot L_O(\lambda, \theta_k; D_k^{val}) + \frac{\delta}{2}\|\lambda\|^2. \tag{7}$$

Because of the minimization condition, we have

$$\sum_{i \in \mathcal{I}} D_\lambda L_O\left(\lambda_\epsilon, \theta_i(\lambda_\epsilon); D_i^{val}\right) + \epsilon \cdot D_\lambda L_O\left(\lambda_\epsilon, \theta_k(\lambda_\epsilon); D_i^{val}\right) + \delta \cdot \lambda_\epsilon = 0 \tag{8}$$

where $D_\lambda L_O\left(\lambda_\epsilon, \theta_i(\lambda_\epsilon); D_i^{val}\right)$ is the total derivative with respect to $\lambda$ defined in Definition B.1, which is

$$D_\lambda L_O\left(\lambda_\epsilon, \theta_i(\lambda_\epsilon); D_i^{val}\right) = \partial_\lambda L_O\left(\lambda_\epsilon, \theta_i(\lambda_\epsilon); D_i^{val}\right) + \frac{d\theta_i(\lambda_\epsilon)}{d\lambda_\epsilon} \cdot \partial_\theta L_O\left(\lambda_\epsilon, \theta_i(\lambda_\epsilon); D_i^{val}\right),$$

where $\frac{d\theta_i(\lambda_\epsilon)}{d\lambda_\epsilon}$ is obtained by

$$\frac{d\theta_i(\lambda_\epsilon)}{d\lambda_\epsilon} = -\partial_\lambda\partial_\theta L_I(\lambda, \theta_i(\lambda_\epsilon); D_i^{tr}) \cdot \partial_\theta\partial_\theta L_I(\lambda, \theta_i(\lambda_\epsilon); D_i^{tr})^{-1}.$$

Then, for $i \in \mathcal{I}$, we perform a Taylor expansion of $D_\lambda L_O\left(\lambda_\epsilon, \theta_i(\lambda_\epsilon); D_i^{val}\right)$ around the point $(\lambda^*, \theta_i(\lambda^*))$, then we have

$$D_\lambda L_O\left(\lambda_\epsilon, \theta_i(\lambda_\epsilon); D_i^{val}\right) \approx D_\lambda L_O(\lambda^*, \theta_i(\lambda^*); D_i^{val}) + D_\lambda D_\lambda L_O(\lambda^*, \theta_i(\lambda^*); D_i^{val}) \cdot (\lambda_\epsilon - \lambda^*) \tag{9}$$

Substituting Equation (9) into (8) and repeating the same process for all the tasks, we have:

$$\delta \cdot \lambda_\epsilon + \sum_{i \in \mathcal{I}} D_\lambda L_O\left(\lambda^*, \theta_i(\lambda^*); D_i^{val}\right) + \epsilon \cdot D_\lambda L_O\left(\lambda^*, \theta_k(\lambda^*); D_k^{val}\right)$$

$$+ \left(\delta \cdot I + \sum_{i \in \mathcal{I}} D_\lambda D_\lambda L_O(\lambda^*, \theta_i(\lambda^*); D_i^{val})\right) \cdot (\lambda_\epsilon - \lambda^*) = 0. \tag{10}$$

The high order terms have been dropped in above equation. From Equation (6), we can derive an optimal condition for $\lambda^*$ as:

$$\sum_{i \in \mathcal{I}} D_\lambda L_O\left(\lambda^*, \theta_i(\lambda^*); D_i^{val}\right) + \delta \cdot \lambda^* = 0.$$

Then the first term in Equation (10) equals 0. By simple algebraic manipulation, we have

$$\lambda_\epsilon - \lambda^* = -\epsilon \cdot \left(\delta \cdot I + \sum_{i \in \mathcal{I}} D_\lambda D_\lambda L_O(\lambda^*, \theta_i(\lambda^*); D_i^{val})\right)^{-1} \cdot D_\lambda L_O\left(\lambda^*, \theta_k(\lambda^*); D_k^{val}\right).$$

Revising the definition of the total Hessian matrix $H_{\lambda,\text{Total}}$ in Definition B.2, the above equation can be rewritten as:

$$\lambda_\epsilon - \lambda^* = -\epsilon \cdot (\delta \cdot I + H_{\lambda,\text{Total}})^{-1} \cdot D_\lambda L_O\left(\lambda^*, \theta_k(\lambda^*); D_k^{val}\right). \tag{11}$$

Letting $\epsilon \to 0$, we can define the task-IF as

$$\text{task-IF}(k) = \left.\frac{d\lambda_\epsilon}{d\epsilon}\right|_{\epsilon=0} = -(\delta \cdot I + H_{\lambda,\text{Total}})^{-1} \cdot D_\lambda L_O\left(\lambda^*, \theta_k(\lambda^*); D_k^{val}\right).$$

When set $\epsilon = -1$, $\lambda_{-1} = \lambda^*_{-k}$, we can use a first-order approximation to estimate $\lambda_{-1}$ as an approximation for $\lambda^*_{-k}$:

$$\lambda^*_{-k} \approx \lambda^* - \cdot\text{para-IF}(k).$$

$\square$

*Remark* B.4. Note that the task influence function depends on the total Hessian matrix. For simplicity, we denote $L_O\left(\lambda, \theta_i(\lambda); D_i^{val}\right)$ as $L_O^i$, then it can be written as

$$H_{\lambda,\text{Total}} = \sum_{i \in \mathcal{I}} \left( \partial_\lambda \partial_\lambda L_O^i + 2 \cdot \frac{\mathrm{d}\theta_i(\lambda)}{\mathrm{d}\lambda} \cdot \partial_{\theta_i} \partial_\lambda L_O^i \right.$$
$$\left. + \frac{\mathrm{d}^2 \theta_i(\lambda)}{\mathrm{d}\lambda^2} \cdot \partial_{\theta_i} L_O^i + \left( \frac{\mathrm{d}\theta_i(\lambda)}{\mathrm{d}\lambda} \right)^2 \cdot \partial_{\theta_i} \partial_{\theta_i} L_O^i \right).$$

*Proof.* From Definition B.2, the total Hessian matrix $H_{\lambda,\text{Total}}$ is defined as:

$$H_{\lambda,\text{Total}} = \sum_{i \in \mathcal{I}} \left( \mathrm{D}_\lambda \mathrm{D}_\lambda L_O \left( \lambda, \theta_i(\lambda); D_i^{val} \right) \right).$$

Then, we can expand the total Hessian matrix $H_{\lambda,\text{Total}}$ as:

$$H_{\lambda,\text{Total}} = \sum_{i \in \mathcal{I}} \left( \mathrm{D}_\lambda \mathrm{D}_\lambda L_O \left( \lambda, \theta_i(\lambda); D_i^{val} \right) \right)$$
$$= \sum_{i \in \mathcal{I}} \mathrm{D}_\lambda \left( \partial_\lambda L_O \left( \lambda, \theta_i(\lambda) \right) + \frac{\mathrm{d}\theta_i(\lambda)}{\mathrm{d}\lambda} \cdot \partial_{\theta_i} L_O \left( \lambda, \theta_i(\lambda) \right) \right)$$
$$= \sum_{i \in \mathcal{I}} \left( \partial_\lambda \partial_\lambda L_O^i + 2 \cdot \frac{\mathrm{d}\theta_i(\lambda)}{\mathrm{d}\lambda} \cdot \partial_{\theta_i} \partial_\lambda L_O^i \right.$$
$$\left. + \frac{\mathrm{d}^2 \theta_i(\lambda)}{\mathrm{d}\lambda^2} \cdot \partial_{\theta_i} L_O^i + \left( \frac{\mathrm{d}\theta_i(\lambda)}{\mathrm{d}\lambda} \right)^2 \cdot \partial_{\theta_i} \partial_{\theta_i} L_O^i \right).$$

$\square$

## B.2 EVALUATING THE INFLUENCE OF INDIVIDUAL INSTANCE

**Theorem B.5** (Instance-IF for Validation Data). *The influence function of the validation data $\tilde{z} = (\tilde{x}, \tilde{y})$ is*

$$\text{instance-IF}(\tilde{z}; \lambda^*, \theta_k(\lambda^*))$$
$$= -\left( \delta \cdot I + H_{\lambda,\text{Total}} \right)^{-1} \nabla \ell(\tilde{z}; \theta_k(\lambda^*)).$$

*After the removal of $\tilde{z}$, the retrained optimal meta parameter $\lambda^*_{-\tilde{z}}$ can be estimated by*

$$\lambda^*_{-\tilde{z}} \approx \lambda^* - \text{instance-IF}(\tilde{z}; \lambda^*, \theta_k(\lambda^*)).$$

*Proof.* Consider evaluating the influence of the data $\tilde{z} = (\tilde{x}, \tilde{y})$ in $\mathcal{D}_k^{val}$ in parameter $\lambda$, the loss function and parameter after the removal is defined as follows:

$$\lambda^*_{-\tilde{z}} = \arg\min_\lambda \sum_{i \in \mathcal{I}} L_O(\lambda, \theta_i(\lambda); \mathcal{D}_k^{val}) - \ell(\tilde{z}; \theta_k(\lambda)) + \frac{\delta}{2} \|\lambda\|^2. \tag{12}$$

To estimate $\lambda^*_{-\tilde{z}}$, firstly, up-weight the term in $L_O$ related to $\tilde{z}$ by $\epsilon$ and obtain the outer level loss function as

$$\sum_{i \in \mathcal{I}} L_O(\lambda, \theta_i(\lambda); \mathcal{D}_i^{val}) + \epsilon \cdot \ell(\tilde{z}; \theta_k(\lambda)) + \frac{\delta}{2} \|\lambda\|^2.$$

Then a set of new parameters $\lambda_\epsilon$ are defined by

$$\lambda_\epsilon = \arg\min_\lambda \sum_{i \in \mathcal{I}} L_O(\lambda, \theta_i(\lambda); \mathcal{D}_i^{val}) + \epsilon \cdot \ell(\tilde{z}; \theta_k(\lambda)) + \frac{\delta}{2} \|\lambda\|^2. \tag{13}$$

The from the minimizing condition in (13), we can obtain:

$$\sum_{i \in \mathcal{I}} \mathrm{D}_\lambda L_O(\lambda_\epsilon, \theta_i(\lambda_\epsilon); \mathcal{D}_i^{val}) + \epsilon \cdot \nabla_\lambda \ell(\tilde{z}; \theta_k(\lambda_\epsilon)) + \delta \cdot \lambda_\epsilon = 0$$

Perform Taylor expansion at $\lambda_\epsilon = \lambda^*$,

$$\sum_{i \in \mathcal{I}} D_\lambda L_O(\lambda^*, \theta_i(\lambda^*); \mathcal{D}_i^{val}) + \epsilon \cdot \nabla_\lambda \ell(\tilde{z}; \theta_k(\lambda^*)) + \delta \cdot (\lambda_\epsilon - \lambda^*)$$

$$+ \sum_{i \in \mathcal{I}} D_\lambda D_\lambda L_O(\lambda^*, \theta_i(\lambda^*); \mathcal{D}_i^{val}) \cdot (\lambda_\epsilon - \lambda^*) \approx 0 \tag{14}$$

From the definition of $\lambda^*$ we can find it as the minimizer of the original outer level loss function $\sum_{i \in \mathcal{I}} L_O(\lambda, \theta_i(\lambda); \mathcal{D}_i^{val}) + \frac{\delta}{2} \|\lambda\|^2$, therefore, the first term in Equation (14) equals 0. Besides, $H_{\lambda,\text{Total}} = \sum_{i \in \mathcal{I}} D_\lambda D_\lambda L_O(\lambda^*, \theta_i(\lambda^*); \mathcal{D}_i^{val})$ is the total Hessian matrix of the outer level loss function. Then we have

$$\lambda_\epsilon - \lambda^* = -\epsilon \cdot (\delta \cdot I + H_{\lambda,\text{Total}})^{-1} \nabla_\lambda \ell(\tilde{z}; \theta_k(\lambda^*))$$

Let $\epsilon \to 0$, we can obtain the definition of the instance-level influence function for validation data as follows:

$$\text{instance-IF}(\tilde{z}; \lambda^*, \theta_k(\lambda^*)) = -(\delta \cdot I + H_{\lambda,\text{Total}})^{-1} \nabla \ell(\tilde{z}; \theta_k(\lambda^*)).$$

When $\epsilon = -1$, the influence of $(\tilde{x}, \tilde{y})$ is fully removed. We can use this instance-IF$(\tilde{z}; \lambda^*, \theta_k(\lambda^*))$ to estimate $\lambda^*_{-\tilde{z}}$ when data $\tilde{z}$ is removed with $\lambda^*$ fixed as

$$\lambda^*_{-\tilde{z}} \approx \lambda^* - \text{instance-IF}(\tilde{z}; \lambda^*, \theta_k(\lambda^*)).$$

$\square$

**Theorem B.6.** *The influence function for the inner level (inner-IF) of $\tilde{z} = (\tilde{x}, \tilde{y})$ from the training set is*

$$\text{inner-IF}(\tilde{z}; \theta_k(\lambda^*)) = -H_{k,in}^{-1} \cdot \nabla_{\theta_k} \ell(\tilde{z}; \theta_k(\lambda^*)),$$

*where $H_{k,in} = \partial_{\theta_k} \partial_{\theta_k} L_I(\lambda^*, \theta_k(\lambda^*); \mathcal{D}_k^{tr})$ is the $k$-th inner-level Hessian matrix. After the removal of $\tilde{z}$, denote $\theta_k^-(\lambda^*)$ as its corresponding retrained task-parameter $\theta_k^-(\lambda^*) = \arg\min_{\theta_i} L_I(\lambda^*, \theta_i; D_i^{tr} \setminus \tilde{z}))$. Then it can be estimated by*

$$\theta_k^-(\lambda^*) \approx \theta_k(\lambda^*) - \text{inner-IF}(\tilde{z}; \theta_k(\lambda^*)).$$

*Proof.* Consider evaluating the influence of the data $\tilde{z} = (\tilde{x}, \tilde{y})$ in $\mathcal{D}_k^{val}$ in parameter $\theta_k(\lambda)$, the loss function and parameter after the removal is defined as follows:

$$\theta_k^-(\lambda) = \arg\min_{\theta_k} L_I(\lambda, \theta_k, \mathcal{D}_k^{tr}) - \ell(\tilde{z}; \theta_k)$$

Revising the definition of $\theta_k(\lambda)$:

$$\theta_k(\lambda) = \arg\min_\theta L_I(\lambda, \theta; \mathcal{D}_k^{tr}) = \sum_{z \in \mathcal{D}_k^{tr}} \ell(z; \theta_k) + \frac{\delta}{2} \|\theta_k - \lambda\|^2.$$

Firstly, up-weight $\ell(\tilde{z}; \theta_k)$ in $L_I$ by $\epsilon$, resulting in the modified loss function. Then, a set of new parameters $\theta_{k,\epsilon}(\lambda)$ are defined by:

$$\theta_{k,\epsilon}(\lambda) = \arg\min_\theta \sum_{z \in \mathcal{D}_k^{tr}} \ell(z; \theta_k) + \frac{\delta}{2} \|\theta_k - \lambda\|^2 + \epsilon \cdot \ell(\tilde{z}; \theta_k) = L_I(\lambda, \theta_k; \mathcal{D}_k^{tr}) + \epsilon \cdot \ell(\tilde{z}; \theta_k). \tag{15}$$

The from the minimizing condition in (15), we can obtain:

$$\nabla_\theta L_I(\lambda, \theta_{k,\epsilon}(\lambda); \mathcal{D}_k^{tr}) + \epsilon \cdot \nabla_\theta \ell(\tilde{z}; \theta_{k,\epsilon}(\lambda)) = 0$$

Perform Taylor expansion at $\theta_{k,\epsilon}(\lambda) = \theta_k(\lambda)$,

$$\nabla_\theta L_I(\lambda, \theta_k; \mathcal{D}_k^{tr}) + \epsilon \cdot \nabla_\theta \ell(\tilde{z}; \theta_k(\lambda)) + \nabla_\theta^2 L_I(\lambda, \theta_k, \mathcal{D}_k^{tr}) \cdot (\theta_{k,\epsilon}(\lambda) - \theta_k(\lambda)) \approx 0$$

From the definition of $\theta_i(\lambda)$ we can find it as the minimizer of $L_I(\mathcal{D}_{tr}^i, \theta_i(\lambda), \lambda)$, therefore, the first term in above equation equals 0. Then we have

$$\theta_{i,\epsilon}(\lambda) - \theta_i(\lambda) = -\epsilon \cdot H_{\theta,i}^{-1} \cdot \nabla_\theta \ell(x, y; \theta_i(\lambda))$$

where $H_{\theta,i} = \nabla^2_{\theta_i} L_I(\mathcal{D}^i_{tr}, \theta_i(\lambda), \lambda)$.

Let $\epsilon \to 0$, we can obtain the definition of the inner-level influence function as follows:

$$\text{inner-IF}(x, y; \theta_k(\lambda)) \triangleq \left. \frac{\mathrm{d}\theta_{k,\epsilon}(\lambda)}{\mathrm{d}\epsilon} \right|_{\epsilon=0} = -H^{-1}_{\theta_k} \cdot \nabla_{\theta_k} \ell(\tilde{x}, \tilde{y}; \theta_k(\lambda))$$

When $\epsilon = -1$, the influence of $(\tilde{x}, \tilde{y})$ is fully removed. We can use this inner-IF$(x, y; \theta_i(\lambda))$ to estimate $\theta^-_i(\lambda)$ when data $(x, y)$ is removed with $\lambda$ fixed as

$$\theta^-_i(\lambda) \approx \theta_i(\lambda) - \text{inner-IF}(x, y; \theta_i(\lambda)) \triangleq \tilde{\theta}_i(\lambda)$$

$\square$

**Theorem B.7.** *If the $k$-th task related parameter changes from $\theta_k$ to $\theta^-_k$, we can estimate the related loss function difference*

$$L_O\left(\left(\lambda, \theta^-_k(\lambda)\right); D^{val}_k\right) - L_O\left(\lambda, \theta_k(\lambda); D^{val}_k\right)$$

*by $P(\lambda, \theta_k(\lambda); \tilde{z})$, which is defined as*

$$-\nabla_\theta L_O\left(\lambda, \theta_k(\lambda); D^{val}_k\right)^{\mathrm{T}} \cdot \text{inner-IF}(\tilde{z}; \theta_k(\lambda)).$$

*Proof.* The loss function difference can be estimated by the first-order Taylor expansion as:

$$L_O\left(\left(\lambda, \theta^-_k(\lambda)\right); D^{val}_k\right) - L_O\left(\lambda, \theta_k(\lambda); D^{val}_k\right)$$

$$= \nabla_\theta L_O\left(\lambda, \theta_k(\lambda); D^{val}_k\right)^{\mathrm{T}} \cdot (\theta^-_k(\lambda) - \theta_k(\lambda))$$

$$\approx -\nabla_\theta L_O\left(\lambda, \theta_k(\lambda); D^{val}_k\right)^{\mathrm{T}} \cdot \text{inner-IF}(\tilde{z}; \theta_k(\lambda)).$$

The second equation is obtained by Theorem B.6. Then the proof is completed. $\square$

**Proposition B.8** (Instance-IF for Training Data). *Let $\zeta \to 0$, we can obtain the influence function of the outer level as follows:*

$$\text{instance-IF}(\tilde{z}; \lambda^*, \theta_k(\lambda^*)) = -H^{-1}_{\lambda, Total} \cdot \mathrm{D}_\lambda P(\lambda^*, \theta_k(\lambda^*); \tilde{z}).$$

*After the removal of $\tilde{z}$, the optimal meta parameter $\lambda^*_{-\tilde{z}}$ can be estimated by*

$$\lambda^*_{-\tilde{z}} \approx \lambda^* + \text{instance-IF}(\tilde{z}; \lambda^*, \theta_k(\lambda^*)).$$

*Proof.* In the following, we will derive the influence of removing $\tilde{z}$ to the meta parameter $\lambda^*$. Follow the settings in the Preliminary part, we expand the form of the outer level loss function as:

$$\sum_{i \in \mathcal{I}} L_O\left(\lambda, \theta_i(\lambda); D^{val}_i\right)$$

$$= \sum_{i \in \mathcal{I}} \sum_{z \in \mathcal{D}^{val}_i} \ell(z; \theta_i(\lambda)) + \frac{\delta}{2} \|\theta_i(\lambda) - \lambda\|^2.$$

If the data $\tilde{z}$ we want to remove is in the training dataset of the $k$-th task, then $\theta_k(\lambda)$ will change to $\theta^-_k(\lambda)$. And we can estimate the loss function difference with the help of Theorem B.7 as follows:

$$L_O\left(\lambda, \theta^-_k(\lambda); D^{val}_k\right) - L_O\left(\lambda, \theta_k(\lambda); D^{val}_k\right) \approx P(\lambda, \theta_k(\lambda); \tilde{z}).$$

Then $P(\lambda, \theta_k(\lambda); \tilde{z})$ is the actual change in the outer-level loss function resulting from the remove of $\tilde{z}$. We add this term into the total loss function and up-weight it by a small $\zeta$ from 0 to 1. When $\zeta = 1$, the influence of remove is fully add into the loss function.

$$\sum_{i \in \mathcal{I}} L_O\left(\lambda, \theta_i(\lambda); D^{val}_i\right) + \zeta \cdot P(\lambda, \theta_k(\lambda); \tilde{z}).$$

And a set of parameter $\lambda^*_\zeta$ is obtained by the minimization process as

$$\lambda^*_\zeta = \arg\min_\lambda \left( \sum_{i \in \mathcal{I}} L_O\left(\lambda, \theta_i(\lambda); D^{val}_i\right) + \zeta \cdot P(\lambda, \theta_k(\lambda); \tilde{z}) \right).$$

Naturally we have the following optimal condition:

$$\sum_{i \in \mathcal{I}} \mathrm{D}_\lambda L_O \left( \lambda_\zeta^*, \theta_i(\lambda_\zeta^*); D_i^{val} \right) + \zeta \cdot \mathrm{D}_\lambda P(\lambda_\zeta^*, \theta_k(\lambda_\zeta^*); \tilde{z}) = 0.$$

Perform a Taylor expand at $\lambda^*$,

$$\sum_{i \in \mathcal{I}} \mathrm{D}_\lambda L_O \left( \lambda^*, \theta_i(\lambda^*); D_i^{val} \right) + \zeta \cdot \mathrm{D}_\lambda P(\lambda^*, \theta_k(\lambda^*); \tilde{z})$$

$$+ \sum_{i \in \mathcal{I}} \mathrm{D}_\lambda \mathrm{D}_\lambda L_O \left( \lambda^*, \theta_i(\lambda^*); D_i^{val} \right) \cdot \left( \lambda_\zeta^* - \lambda^* \right) \approx 0.$$

Noting here $\lambda^*$ is the minimizer for the original loss function $\sum_{i \in \mathcal{I}} L_O \left( \lambda, \theta_i(\lambda); D_i^{val} \right)$, therefore the first term equal 0, then we have

$$\lambda_\zeta^* - \lambda^* = -\zeta \cdot H_{\lambda,\mathrm{Total}}^{-1} \cdot \mathrm{D}_\lambda P(\lambda^*, \theta_k(\lambda^*); \tilde{z}).$$

where $H_{\lambda,\mathrm{Total}} = \sum_{i \in \mathcal{I}} \mathrm{D}_\lambda \mathrm{D}_\lambda L_O \left( \lambda^*, \theta_i(\lambda^*); D_i^{val} \right)$. Let $\zeta \to 0$, we can obtain the definition of the instance level influence function as follows:

$$\text{instance-IF}(\tilde{z}; \lambda^*, \theta_k(\lambda^*)) \triangleq \left. \frac{\mathrm{d}\lambda_\zeta^*}{\mathrm{d}\zeta} \right|_{\zeta=0} = -H_{\lambda,\mathrm{Total}}^{-1} \cdot \mathrm{D}_\lambda P(\lambda^*, \theta_k(\lambda^*); \tilde{z})$$

When $\zeta = 1$, the influence of removing $\tilde{z}$ is fully added into the outer level loss function. We can use this instance-IF$(\tilde{z}; \lambda^*, \theta_k(\lambda^*))$ to estimate the meta parameter $\lambda_-^*$ when data $\tilde{z}$ is removed as follows:

$$\lambda_-^* \approx \lambda^* + \text{instance-IF}(\tilde{z}; \lambda^*, \theta_k(\lambda^*)).$$

$\square$

### B.3 Applications of Influence Functions

**iHVP Acceleration via EK-FAC** We utilize EK-FAC to perform an approximation for $H_{i,\mathrm{in}}$, and then compute the inversion efficiently. Revising the definition of $H_{i,\mathrm{in}}$:

$$H_{i,\mathrm{in}} = \partial_\theta \partial_\theta L_I(\lambda^*, \theta_i(\lambda^*); \mathcal{D}_i^{tr}).$$

The EK-FAC method relies on two approximations of the Fisher information matrix $G_\theta$, which is equivalent to $H_{i,\mathrm{in}}$ when the loss function $\ell$ in $L_I(\lambda^*, \theta_i(\lambda^*); \mathcal{D}_i^{tr})$ is specified as cross-entropy. Firstly, assume that the derivatives of the weights in different layers are uncorrelated, which implies that $H_{i,\mathrm{in}}$ has a block-diagonal structure.

Suppose model $\hat{f}_\theta$ can be denoted by $\hat{f}_\theta(x) = f_{\theta_L} \circ \cdots \circ f_{\theta_1}(x)$. We fold the bias into the weights and vectorize the parameters in the $l$-th layer into a vector $\theta_l \in \mathbb{R}^{d_l}$, $d_l \in \mathbb{N}$ is the number of $l$-th layer parameters.

Then $G_\theta$ can be replaced by $(G_1(\theta), \cdots, G_L(\theta))$, where

$$G_l(\theta) \triangleq n^{-1} \sum_{i=1}^n \nabla_{\theta_l} L_I(\lambda^*, \theta_i(\lambda^*); \mathcal{D}_i^{tr}) \nabla_{\theta_l} L_I(\lambda^*, \theta_i(\lambda^*); \mathcal{D}_i^{tr})^{\mathrm{T}}.$$

Denote $h_l$, $o_l$ as the output and pre-activated output of $l$-th layer. Define $\Omega_{l-1} = \hat{G}_l(\theta) \triangleq \frac{1}{n} \sum_{i=1}^n h_{l-1}(x_i) h_{l-1}(x_i)^T \otimes \frac{1}{n} \sum_{i=1}^n \nabla_{o_l} \ell_i \nabla_{o_l} \ell_i^T$. Then $G_l(\theta)$ can be approximated by

$$G_l(\theta) \approx \Omega_{l-1} \otimes \Gamma_l.$$

Furthermore, in order to accelerate transpose operation and introduce the damping term, perform eigenvalue decomposition of matrix $\Omega_{l-1}$ and $\Gamma_l$ and obtain the corresponding decomposition results as $Q_\Omega \Lambda_\Omega Q_\Omega^\top$ and $Q_\Gamma \Lambda_\Gamma Q_\Gamma^\top$. Then the inverse of $\hat{H}_l(\theta)$ can be obtained by

$$\hat{H}_l(\theta)^{-1} \approx \left( \hat{G}_l(\hat{g}) + \lambda_l I_{d_l} \right)^{-1} = \left( Q_{\Omega_{l-1}} \otimes Q_{\Gamma_l} \right) \left( \Lambda_{\Omega_{l-1}} \otimes \Lambda_{\Gamma_l} + \lambda_l I_{d_l} \right)^{-1} \left( Q_{\Omega_{l-1}} \otimes Q_{\Gamma_l} \right)^{\mathrm{T}}. \tag{16}$$

Besides, George et al. (2018) proposed a new method that corrects the error in equation 16 which sets the $i$-th diagonal element of $\Lambda_{\Omega_{l-1}} \otimes \Lambda_{\Gamma_l}$ as $\Lambda_{ii}^* = n^{-1} \sum_{j=1}^n \left( \left( Q_{\Omega_{l-1}} \otimes Q_{\Gamma_l} \right) \nabla_{\theta_l} \ell_j \right)_i^2$.

**Definition B.9** (Evaluation Function for Meta Learning). Given a new task specific dataset $D_{new} = D_{new}^{tr} \cup D_{new}^{val}$, the few-shot learned parameter $\theta'$ for the task is obtained by minimizing the loss $L_I(\lambda^*, \theta; D_{new}^{tr})$. We can evaluate the performance of the meta parameter $\lambda^*$ based on the loss of $\theta'$ on the validation dataset, expressed as $\sum_{z \in D_{new}^{val}} \ell(z; \theta')$.

Based on this metric, we propose a calculation method for the task and instance Influence Score, defined as $\mathrm{IS} = \sum_{z \in D_{new}^{val}} \nabla_{\theta'} \ell(z; \theta') \cdot \mathrm{IF}$.

*Proof.* Assume we consider the contribution of a set of data $D$, define the initial optimal meta parameter as $\lambda^*$, and the retrained optimal meta parameter after removing the data $D$ as $\lambda^*_{-D}$. We define the evaluation metric as the test loss difference:

$$\sum_{z \in D_{new}^{val}} \ell(z; \theta'(\lambda^*)) - \sum_{z \in D_{new}^{val}} \ell(z; \theta'(\lambda^*_{-D}))$$

We can estimate this difference by a Taylor expansion as follows:

$$\sum_{z \in D_{new}^{val}} \ell(z; \theta'(\lambda^*)) - \sum_{z \in D_{new}^{val}} \ell(z; \theta'(\lambda^*_{-D}))$$
$$= \sum_{z \in D_{new}^{val}} \nabla_{\theta'} \ell(z; \theta'(\lambda^*)) \cdot (\theta'(\lambda^*) - \theta'(\lambda^*_{-D}))$$
$$= \sum_{z \in D_{new}^{val}} \nabla_{\theta'} \ell(z; \theta'(\lambda^*)) \cdot \frac{\mathrm{d}\theta'(\lambda^*)}{\mathrm{d}\lambda^*} \left(\lambda^* - \lambda^*_{-D}\right).$$

To make our method more efficient, we neglect the influence in $\frac{\mathrm{d}\theta'(\lambda^*)}{\mathrm{d}\lambda^*}$ and directly estimate the loss difference by

$$\sum_{z \in D_{new}^{val}} \nabla_{\theta'} \ell(z; \theta'(\lambda^*)) \cdot \left(\lambda^* - \lambda^*_{-D}\right),$$

in which $\lambda^* - \lambda^*_{-D}$ can be estimated by our proposed IF. Then we define the Influence Score(IS) as $\mathrm{IS} = \sum_{z \in D_{new}^{val}} \nabla_{\theta'} \ell(z; \theta'(\lambda^*)) \cdot \mathrm{IF}$, and use it to evaluate the influence of the data $D$. $\square$

## C  FULL ALGORITHMS

---
**Algorithm 1** Task-IF
---
1: **Input:**
   **Data:** Training Dataset $\mathcal{D} = \cup_{i \in \mathcal{I}} \{\mathcal{D}_i^{tr} \cup \mathcal{D}_i^{val}\}$, task $k$ to be evaluated.
   **Parameter:** The learned meta parameter $\lambda^*$, the task-related parameters $\theta_i$ for $i \in \mathcal{I}$.
2: Subsample a task subset $\mathcal{I}_S$. For the detailed methods, there are multiple choices, including random sampling, difficulty-based sampling, uncertainty-based sampling.
3: Compute the **total gradient** by Algorithm 3.
4: Use **Neumann Series Approximation method** to compute the task-IF by Algorithm 4.
5: Approximate the model retained when the $k$-th task is removed: $\lambda^*_- = \lambda^* - \text{task-IF}$
6: **Return:** task-IF, $\lambda^*_-$.

---

---

**Algorithm 2** Direct IF

---

1: **Input:**
   **Data:** Training Dataset $\mathcal{D} = \cup_{i \in \mathcal{I}} \{\mathcal{D}_i^{tr} \cup \mathcal{D}_i^{val}\}$, task $k$ to be evaluated.
   **Parameter:** The learned meta-parameter $\lambda^*$, the task-related parameters $\theta_i$ for $i \in \mathcal{I}$.
2: Subsample a task subset $\mathcal{I}_S$. For the detailed methods, there are multiple choices, including random sampling, difficulty-based sampling, uncertainty-based sampling.
3: Compute the partial gradient of the $k$-th outer loss function as

$$T^k = \partial_\lambda L_O(\lambda^*, \theta_k; \mathcal{D}_k^{val}).$$

4: Compute the direct-IF($k$) based on $T^k$:

$$\text{direct-IF} = -\left(\sum_{i \in \mathcal{I}} \partial_\lambda \partial_\lambda L_O(\lambda^*, \theta_i; \mathcal{D}_i^{val}) + \delta \cdot I\right)^{-1} \cdot T^k.$$

5: Approximate the model retained when the $k$-th task is removed: $\lambda_-^* = \lambda^* - \text{direct-IF}$.
6: **Return:** direct-IF($k$), $\lambda_-^*$.

---

**Algorithm 3** Total Gradient Computation for Task

---

1: **Input:**
   **Data:** Training Dataset $\mathcal{D} = \cup_{i \in \mathcal{I}} \{\mathcal{D}_i^{tr} \cup \mathcal{D}_i^{val}\}$, task $k$ to be evaluated.
   **Parameter:** the learned meta-parameter $\lambda^*$, the task-related parameters $\theta_i$ for $i \in \mathcal{I}$.
2: Use EK-FAC to compute the middle gradient $M$ as

$$M_k = \partial_\lambda \partial_\theta L_I\left(\lambda^*, \theta_k; \mathcal{D}_k^{tr}\right) \cdot \partial_\theta \partial_\theta^{-1} L_I\left(\lambda^*, \theta_k; \mathcal{D}_k^{tr}\right) \cdot \partial_\theta L_O(\lambda^*, \theta_k; \mathcal{D}_k^{val}).$$

3: Compute the total derivative of the $k$-th outer loss function with respect to $\lambda^*$ as $T^k$:

$$T^k = \partial_\lambda L_O(\lambda^*, \theta_k; \mathcal{D}_k^{val}) - M_k$$

.
4: **Return:** Total Gradient of the $k$-th task $T_k$.

---

**Algorithm 4** Neumann Expansion Based Total Hessian Computation

---

1: **Input:**
   **Data:** Training Dataset $\mathcal{D} = \cup_{i \in \mathcal{I}} \{\mathcal{D}_i^{tr} \cup \mathcal{D}_i^{val}\}$, task $k$ to be evaluated.
   **Parameter:** The learned meta-parameter $\lambda^*$, the task-related parameters $\theta_i$ for $i \in \mathcal{I}$.
   **Task Gradient:** $T$
2: $j \leftarrow 1$
   $T \leftarrow I_0$
3: **if** $\|I_j - I_{j-1}\|_1 > \zeta$ **then**
4:    **for** $i \in \mathcal{I}$ **do**
5:      The total Hessian $H^i$ related to the $i$-th task is defined as

$$\partial_\lambda \partial_\lambda L_O\left(\lambda^*, \theta_i; \mathcal{D}_i^{val}\right) + \|\partial_\theta L_O\left(\lambda^*, \theta_i; \mathcal{D}_i^{val}\right)\|_1 \cdot I$$
$$- 2 \cdot \partial_\lambda \partial_\theta L_I\left(\lambda^*, \theta_i; \mathcal{D}_i^{tr}\right) \cdot \partial_\theta \partial_\theta^{-1} L_I\left(\lambda^*, \theta_i; \mathcal{D}_i^{tr}\right) \cdot \partial_\theta \partial_\lambda L_O\left(\lambda^*, \theta_i; \mathcal{D}_i^{val}\right),$$

     where $I$ is the identity matrix.
6:    **end for**
7:    Use EK-FAC to compute the Hessian-vector product $S_j \triangleq \sum_{i \in \mathcal{I}} H^i \cdot I_j$.
8:    Compute $I_{j+1}$ by
$$I_{j+1} = I_j - \delta \cdot I_j + S_j + T$$

9:    $j \leftarrow j + 1$
10: **end if**
11: **Return:** IF $= I_j$.

---

---

**Algorithm 5** Instance-level IF for Validation Data

---

1: **Input:**
   **Data:** Training Dataset $\mathcal{D} = \cup_{i \in \mathcal{I}} \{\mathcal{D}_i^{tr} \cup \mathcal{D}_i^{val}\}$, data $(\tilde{x}, \tilde{y})$ to be evaluated, the corresponding task $k$,
   **Parameter:** The learned meta-parameter $\lambda^*$, the task-related parameters $\theta_i$ for $i \in \mathcal{I}$.
2: Use EK-FAC to compute the middle gradient $M$ as

$$M = \partial_\lambda \partial_\theta L_I \left(\lambda^*, \theta_k; \mathcal{D}_k^{tr}\right) \cdot \partial_\theta \partial_\theta^{-1} L_I \left(\lambda^*, \theta_k; \mathcal{D}_k^{tr}\right) \cdot \partial_\theta \ell(\tilde{x}, \tilde{y}; \theta_k).$$

3: Compute the total derivative of the $k$-th outer loss function with respect to $\lambda^*$ as $T$:

$$T = \partial_\lambda \ell(\tilde{x}, \tilde{y}; \theta_k) - M.$$

4: Take $T$ as the gradient to be the input, use **Neumann Series Approximation method** to compute the instance-IF for validation data by Algorithm 4.
5: Approximate the model retained when the $k$-th task is removed: $\lambda_-^* = \lambda^* -$ instance-IF
6: **Return:** instance-IF, $\lambda_-^*$.

---

---

**Algorithm 6** Instance-IF for Training Data

---

1: **Input:**
   **Data:** Training Dataset $\mathcal{D} = \cup_{i \in \mathcal{I}} \{\mathcal{D}_i^{tr} \cup \mathcal{D}_i^{val}\}$, data $(\tilde{x}, \tilde{y})$ to be evaluated, the corresponding task $k$,
   **Parameter:** The learned meta-parameter $\lambda^*$, the task-related parameters $\theta_i$ for $i \in \mathcal{I}$.
2: Compute inner-IF for the $k$-th task as:

$$\text{inner-IF} = -\partial_\theta \partial_\theta^{-1} L_I \left(\lambda^*, \theta_k; \mathcal{D}_k^{tr}\right) \cdot \partial_\theta \ell(\tilde{x}, \tilde{y}; \theta_k).$$

3: Compute the outer influence term $P_k$ as

$$P_k = -\partial_\theta L_O \left(\lambda^*, \theta_k; D_k^{val}\right)^{\mathrm{T}} \cdot \text{inner-IF}.$$

4: Take $T$ as the gradient to be the input, use **Neumann Series Approximation method** to compute the instance-IF for validation data by Algorithm 4.
5: Approximate the model retained when the data is removed: $\lambda_-^* = \lambda^* +$ instance-IF
6: **Return:** instance-IF, $\lambda_-^*$.

---

## D    EXPERIMENT DETAILS

**Dataset.** Omniglot features 1,623 handwritten characters from 50 alphabets. MNIST is a handwritten digit dataset with 60,000 training examples and 10,000 test examples. Mini-ImageNet contains 50,000 training images and 10,000 test images across 100 classes. FC100 is a few-shot learning benchmark with 60,000 CIFAR-100 images, divided into 64 training, 16 validation, and 20 test classes for systematic cross-category evaluation.

**Implementation Details.** Our experiments used an Intel Xeon CPU and a GTX 1080 ti GPU. For task-level evaluation, we train the MAML model, randomly remove some tasks, and then use direct IF and task-IF to update the model. For instance level, we randomly remove $40\%$ sample for 20 tasks. During the training phase, we update the model for 5000 iterations and randomly sample 10,000 tasks for the omniglot and MINI-Imagenet datasets and 252 tasks for the MNIST dataset. We repeat the experiments 5 times with different random seeds.

## E    ADDITIONAL VISUALIZATION

## F    LIMITATION

In this work, we employ influence function to efficiently approximate the performance change that would occur if a particular data instance or task were removed and the model were retrained (leave-one-out retraining). It is important to note that our method still fundamentally relies on estimating retraining outcomes rather than performing actual retraining. As a result, while we provides a highly efficient and effective approximation in practice, it may not capture all nuances of true retraining, and potential discrepancies may exist between the estimated and actual data values. However, experimental results show that the influence function-based approximations closely reflect the outcomes of actual retraining, making our data attribution method both practical and effective in real-world meta-learning applications.

**Mini-ImageNet Dataset**

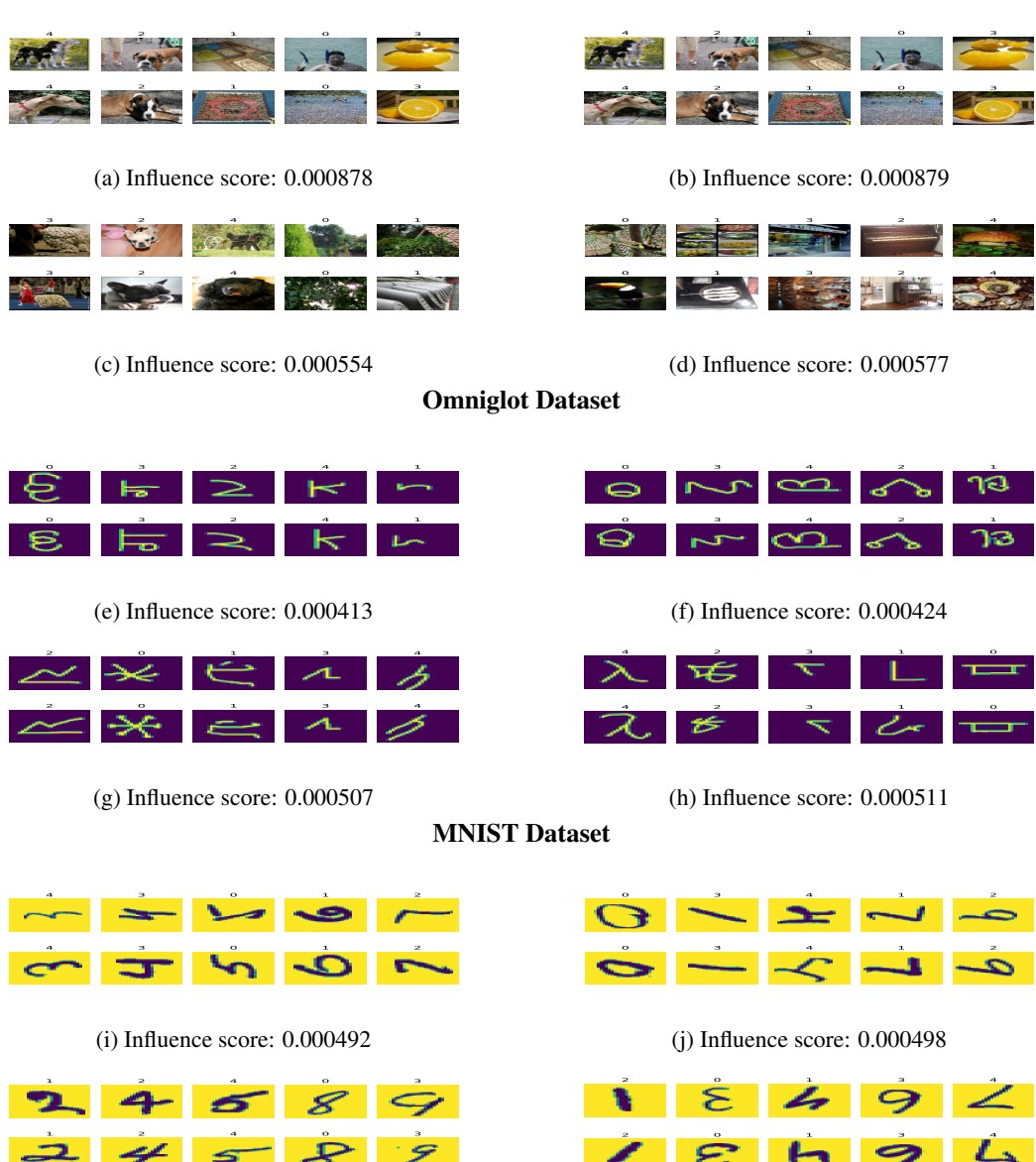

(a) Influence score: 0.000878

(b) Influence score: 0.000879

(c) Influence score: 0.000554

(d) Influence score: 0.000577

**Omniglot Dataset**

(e) Influence score: 0.000413

(f) Influence score: 0.000424

(g) Influence score: 0.000507

(h) Influence score: 0.000511

**MNIST Dataset**

(i) Influence score: 0.000492

(j) Influence score: 0.000498

(k) Influence score: 0.000537

(l) Influence score: 0.000557

