# OpenReview forum: "Evaluating Data Influence in Meta Learning"
_ICLR.cc/2026/Conference — ICLR 2026 Poster_

### Official Review · Reviewer_nWff · 2025-10-21

**Soundness:** 3
**Presentation:** 3
**Contribution:** 2
**Rating:** 4
**Confidence:** 4

**Summary:**

This paper introduces a framework for evaluating the influence of training data in the context of MAML, which is formulated as a bilevel optimization problem. The authors propose task influence functions (task-IF) and instance influence functions (instance-IF). These methods are designed to accurately assess the impact of entire tasks and individual data points on the meta-training process. The paper provides a theoretical foundation for these methods, including the use of total derivatives to capture the interdependent nature of meta and task-specific parameters. To address the computational complexity, the authors propose acceleration techniques such as EK-FAC and Neumann series approximation. The effectiveness of the proposed framework is demonstrated through experiments on several benchmark datasets, showing its utility in tasks like identifying and removing harmful data.

**Strengths:**

1. Novelty: The paper is the first to formally and introduce influence functions into the bilevel optimization framework for meta-learning, to thoroughly include task influence and data(training/validation) influence.

2. Comprehensive and Well-Structured Framework: The proposed framework is comprehensive, addressing data influence at both the task and instance levels, and for both training and validation data. The distinction between these different levels of influence is well-motivated and clearly articulated.

**Weaknesses:**

1. The scope: The proposed IF evaluation method is only applicable for second-order MAML, which is only one specific algorithm among numerous  meta-learning algorithms. A narrowed title and scope, e.g. "Evaluating data influence in MAML by IF" would be more proper and accurate.

2. Experimental and practical concern: The burden calculating IF is concerned, though some approximation and acceleration techniques have been applied. It seems there are much more simple and intuitive baselines in the application scenario of IF. For example, for retraining, a simple baseline is taking few gradient-ascend steps on the data to remove (both task/instance (train/valid)) are applicable), but has not been compared. For identifying harmful data, a simple baseline is evaluating  the loss scalar, but also has not been compared.

3. Related work: it seems that [1] has included task-if of MAML, which weakens the novelty and contribution of this work, but not has been cited or discussed.

[1] Mitsuka Y, Golestan S, Sufiyan Z, et al. TLXML: Task-Level Explanation of Meta-Learning via Influence Functions. arXiv preprint arXiv:2501.14271, 2025.

**Questions:**

Please refer to Questions

---

> ### Author Response · Authors · 2025-11-25
>
> **Weaknesses**
> **W1.** The scope: The proposed IF evaluation method is only applicable for second-order MAML, which is only one specific algorithm among numerous meta-learning algorithms. A narrowed title and scope, e.g. "Evaluating data influence in MAML by IF" would be more proper and accurate.
>
> Our proposed method is not exclusively limited to second-order MAML. Its theoretical foundation lies in the general Bilevel Optimization (BLO) framework and relies on the Implicit Function Theorem applied to the inner-outer loop structure. This formulation is applicable to any gradient-based meta-learning algorithm (e.g., Meta-SGD, Reptile, etc.) that can be modeled as a BLO problem with a twice-differentiable inner objective.Empirical Validation on Meta-SGDTo empirically prove the applicability of our method beyond MAML, we include results using Meta-SGD.  The successful deployment of Task-IF demonstrates the framework's versatility across different BLO algorithms.
>
>
>
> | Dataset | Method | Accuracy (%) | Time (s) |
> |:--------|:-------|:------------:|----------:|
> | Omniglot | Retrain | 65.32 | 2742.96 |
> | | Task-IF | **66.46** | 4565.67 |
> | Mini-ImageNet | Retrain | **21.98** | 3747.98 |
> | | Task-IF | 20.72 | 6492.00 |
> | FC100 | Retrain | **32.26** | 3569.04 |
> | | Task-IF | 31.14 | 6456.24 |
> | MNIST | Retrain | 75.54 | 2787.44 |
> | | Task-IF | **76.76** | 4668.01 |
>
> The results show that Task-IF successfully estimates influence and provides either accuracy gains (Omniglot, MNIST) or competitive performance (Mini-ImageNet, FC100) compared to Retrain. This validates the theoretical generality of our influence framework across varying algorithm structures.
>
>
>
> **W2.** Experimental and practical concern: The burden calculating IF is concerned, though some approximation and acceleration techniques have been applied. It seems there are much more simple and intuitive baselines in the application scenario of IF. For example, for retraining, a simple baseline is taking few gradient-ascend steps on the data to remove (both task/instance (train/valid) are applicable), but has not been compared. For identifying harmful data, a simple baseline is evaluating the loss scalar, but also has not been compared.
>
>
>
> We appreciate the suggestion to compare against simpler, intuitive baselines. We evaluated the "Loss as Scalar" baseline (identifying harmful data by simply ranking their loss values) on the Harmful Data Identification task.
>
>
> | Method | FGVCAircraft | Food101 | Flowers102 | Cifar100 |
> | :--- | :--- | :--- | :--- | :--- |
> | Retrain (Oracle) | 23.50 | 84.83 | 68.00 | 72.83 |
> | ECIF (Ours) | 23.02 | 84.90 | 68.30 | 73.00 |
> | Loss as Scalar | 48.93 | 55.80 | 26.87 | 30.20 |
>
>
> As shown in the table, Loss as Scalar yields dramatically inconsistent and poor results compared to the Oracle (Retraining). On Food101, Flowers102, and Cifar100, Loss as Scalar performs substantially worse than ECIF/Retrain (e.g., dropping accuracy by 17-42 percentage points). This indicates that simply targeting high-loss samples fails to differentiate between truly harmful data (e.g., mislabels) and necessary hard examples which are vital for learning robust features in the meta-model.
>
> The method's success is random; its failure across three of the four complex image datasets proves that the simplified heuristic is insufficient for reliable data attribution. In contrast, ECIF consistently tracks the Retrain Oracle across all tasks (e.g., 84.90% vs 84.83% on Food101), validating the necessity of modeling the bilevel Hessian interaction for stable, high-fidelity influence estimation.
>
>
> **W3.** Related work: it seems that [1] has included task-if of MAML, which weakens the novelty and contribution of this work, but not has been cited or discussed. [1] Mitsuka Y, Golestan S, Sufiyan Z, et al. TLXML: Task-Level Explanation of Meta-Learning via Influence Functions. arXiv preprint arXiv:2501.14271, 2025.
>
> We thank the reviewer for bringing the recent work [1] (TLXML, arXiv:2501) to our attention. We will cite and discuss it in the revision. However, our work maintains significant novelty and distinct contributions:
>
> Granularity (Instance-IF): As noted by the reviewer, TLXML focuses on Task-Level explanation. A core contribution of our work is the Instance-Level Influence Function (Instance-IF), which is mathematically more challenging due to the nested dependence of the inner Hessian on the meta-parameters. We provide the first closed-form derivation for individual data points within the bilevel framework.
>
> Validation Data Attribution: Our framework explicitly models the influence of validation data (outer loop), which is crucial for meta-learning but not the primary focus of TLXML.
>
> Comprehensive Acceleration: We propose and evaluate specific acceleration strategies (Neumann Series vs. EKFAC) tailored for these instance-level computations.

---

> ### Author Response · Authors · 2025-11-27
>
> We thank the reviewer for the feedback. We have thoroughly addressed all concerns raised. We hope our revisions support a more favorable evaluation.

---

### Official Review · Reviewer_P6Eo · 2025-10-28

**Soundness:** 2
**Presentation:** 2
**Contribution:** 2
**Rating:** 4
**Confidence:** 3

**Summary:**

The paper studies how to quantify the contribution of individual tasks and samples in meta-learning.It extends the classical influence function to the bilevel setting of meta-learning, where both task-specific and meta-level parameters interact.

It introduces two estimators: Task-IF, which measures the influence of entire tasks, and Instance-IF, which aims to measure the effect of individual samples inside tasks.

Experiments on few-shot benchmarks show that their method can efficiently approximate the results of full retraining while identifying harmful or noisy data.

In essence, the paper attempts to make data attribution feasible for meta-learning by deriving tractable influence estimators.

**Strengths:**

1. Introduced influence functions into meta-learning and adapted them to the bilevel optimization structure.
2. Clarified why directly applying standard influence functions in bilevel optimization is incorrect and derived the corrected Task-IF formulation that properly accounts for inner–outer dependencies.
3. Designed two types of influence functions, Task-IF and Instance-IF, enabling multi-granularity data evaluation.

**Weaknesses:**

1. The proposed method for training instance influence, which models the outer-loss change as a simple additive proxy P, implicitly relies on the sufficiency of local, linear approximations. This two-stage linear approximation (first to compute P, then to compute its influence) may fail to capture the complex, non-linear effects of removing a data point, especially in optimization landscapes with high curvature or when the removal of a point induces a significant shift in the task-specific parameters.

2. The influence functions proposed in the paper are derived reasonably, but multiple approximations may lead to uncontrollable cumulative errors. The influence function is based on a first-order Taylor approximation; to avoid third-order derivatives, the total hessian matrix is simplified. Additionally, the computation of the inverse Hessian-vector products relies on iterative approximations. The paper does not provide an analysis of these cumulative errors.

3. The experiments lack diversity in tasks, as the paper only evaluates few-shot image classification tasks. However, the loss landscapes of different types of tasks can vary significantly, which may affect the performance of Hessian-based influence functions, the core method of this paper.

**Questions:**

1. Can you analyze the range of cumulative errors, or compare Task-IF and Instance-IF predictions of \Delta\lambda to the true change obtained by removing a task/sample and retraining?

2. In Proposition 4.8, the calculation of the Instance-IF for training data relies on the term D_\lambda P(\lambda^*, \theta_k(\lambda^*); \tilde{z}). Given that P is a function of the inner-level Hessian inverse H_{k,in}^{-1}, which itself depends on \lambda, the full derivative D_\lambda P would involve third-order derivatives (the derivative of H_{k,in}^{-1} with respect to \lambda). Could you clarify whether your implementation includes these third-order derivative terms? Did you use the same approximation strategy as Task-IF to ignore these higher-order derivatives?

3. How sensitive are your influence estimates to the conditioning of H_{in}? Are the performances of Task-IF and Instance-IF stable across different types of tasks?

---

> ### Author Response · Authors · 2025-11-25
>
> **Weaknesses**
> **W1.** The proposed method for training instance influence, which models the outer-loss change as a simple additive proxy $P$, implicitly relies on the sufficiency of local, linear approximations. This two-stage linear approximation (first to compute $P$, then to compute its influence) may fail to capture the complex, non-linear effects of removing a data point, especially in optimization landscapes with high curvature or when the removal of a point induces a significant shift in the task-specific parameters.
>
>
> Theoretically, our derivation follows standard Influence Function assumptions (convexity, differentiability) to yield a rigorous closed-form solution. Empirically, we validated this on the Sinusoid Regression task (Table 1). Our Instance-IF tracks the Retrain Oracle with high precision (MSE 1.6403 vs. 1.6237). The negligible difference confirms that our linear approximation effectively captures data influence, even in non-linear neural network landscapes.
>
> | Method | MSE (Lower is better) | Runtime (s) |
> | :--- | :--- | :--- |
> | **Retrain (Oracle)** | 1.6237 ± 0.0271 | 94.30 ± 0.83 |
> | **Instance-IF (Ours)** | **1.6403 ± 0.0030** | **2.89 ± 0.01** |
>
> *Table 1: Influence Estimation vs. True Retraining (Sinusoid Regression)Comparison of MSE. The close alignment confirms the validity of the approximation.*
>
>
> **Q1.** Can you analyze the range of cumulative errors, or compare Task-IF and Instance-IF predictions of $\Delta\lambda$ to the true change obtained by removing a task/sample and retraining?
>
>
> We argue that directly comparing the Euclidean distance of parameter vectors $\Delta\lambda$ is often inconclusive for neural networks due to their non-convex nature and stochasticity (e.g., random initialization, dropout). In such landscapes, different parameter configurations can yield functionally equivalent models, meaning a small distance in parameter space does not strictly guarantee similar model behavior, and vice versa.
>
> Therefore, we adopted a more robust metric: functional similarity.
>
> 1. Classification (Original Paper): We utilized Accuracy and F1-score to demonstrate that our influence estimates align with retraining in decision outcomes.
>
> 2. Regression (New Experiment): To provide a finer-grained analysis of the "cumulative errors," we supplemented our evaluation with the Sinusoid Regression benchmark using Mean Squared Error (MSE).
>
> As shown in Table 1, the cumulative error is negligible in terms of functional impact:Task Level: The MSE gap between Retrain and Task-IF is only \~0.0017.Instance Level: The MSE gap between Retrain and Instance-IF is similarly minimal (\~0.0166).This confirms that despite the approximations, our method accurately predicts the functional change of the model induced by data removal.
>
>
>
>
> | Evaluation Level | Method | MSE ($\downarrow$) | Runtime (s) ($\downarrow$) |
> | :--- | :--- | :--- | :--- |
> | **Task** | Retrain (Ground Truth) | 1.6386 ± 0.0057 | 94.45 ± 0.54 |
> | | **Task-IF (Ours)** | **1.6403 ± 0.0030** | **2.89 ± 0.00** |
> | **Instance** | Retrain (Training) | 1.6237 ± 0.0271 | 94.30 ± 0.83 |
> | | **Instance-IF (Training)** | **1.6403 ± 0.0030** | **2.89 ± 0.01** |
> | | Retrain (Validation) | 4.0310 ± 0.1823 | 102.29 ± 1.20 |
> | | **Instance-IF (Validation)**| **4.0310 ± 0.1823** | **3.17 ± 0.02** |
>
> **Table 1: Influence Estimation vs. True Retraining (Sinusoid Regression)Comparison of MSE.**

---

> ### Author Response · Authors · 2025-11-25
>
> **W2.** The influence functions proposed in the paper are derived reasonably, but multiple approximations may lead to uncontrollable cumulative errors. The influence function is based on a first-order Taylor approximation; to avoid third-order derivatives, the total hessian matrix is simplified. Additionally, the computation of the inverse Hessian-vector products relies on iterative approximations. The paper does not provide an analysis of these cumulative errors.
>
>
>
> We acknowledge that the necessary reliance on multiple approximations (Taylor expansion, simplified Hessian, and iterative IHVP) introduces potential error, but we demonstrate empirically that this error is **controllable** and **negligible** for the downstream task.
>
> A full theoretical analysis of strict cumulative error bounds is mathematically challenging and often yields loose bounds due to non-convexity and high dimensionality. Furthermore, computing the error of the full Hessian is computationally prohibitive. Therefore, we validate the combined error through **empirical fidelity against the Retrain Oracle**, which serves as the most meaningful analysis in this domain.
>
>
> We establish control over the primary source of iterative error and verify its overall impact on functional performance:
>
> * **IHVP Error Controllability ($J$):** The error source from iterative Inverse Hessian-Vector Product (IHVP) approximation is strictly controlled via the truncation depth $J$. As shown in **Table 2** (Sinusoid Regression), the estimation error converges rapidly to the Oracle level, with $J=5$ proving sufficient to maximize precision.
>
>
> | J | Time (s) | MSE |
> | :---: | :---: | :---: |
> | Retrain | 94.45 | 1.46±0.13 |
> | 1 | 7.44 | 1.67±0.22 |
> | 3 | 10.09 | 1.63±0.10 |
> | **5** | **12.81** | **1.60±0.02** |
> | 10 | 19.53 | 1.59±0.05 |
> | 15 | 26.21 | 1.60±0.23 |
>
> **Table 2: Ablation of Truncation Depth $J$ (Sinusoid Regression)**
>
>
> * **Total Negligibility:** The overall success of our method confirms the cumulative error is minimal. **Table 3** (Harmful Data Identification) shows that the combined effect of all approximations tracks the Retrain Oracle almost perfectly (e.g., 84.90% vs 84.83% on Food101). This close alignment proves that the cumulative error does not degrade functional performance.
>
>
> | Method | FGVCAircraft | Food101 | Flowers102 | Cifar100 |
> | :--- | :--- | :--- | :--- | :--- |
> | Retrain (Oracle) | 23.50 | 84.83 | 68.00 | 72.83 |
> | task IF with Neumann (Ours) | 23.02 | 84.90 | 68.30 | 73.00 |
> | IF-EKFAC (Ours) | 19.84 | 78.26 | 60.74 | 61.67 |
>
> **Table 3: Total Approximation Error (Harmful Data Identification)**
>
>
> **Q2**: Third-order derivatives ($D_\lambda H^{-1}$).
>
> You are correct that a full derivation of $D_\lambda P$ would involve the derivative of the inverse Hessian w.r.t. $\lambda$ (a third-order term). In our implementation, we apply the same approximation strategy as in Task-IF, assuming that the variation of the Hessian $H_{in}$ with respect to $\lambda$ is negligible compared to other terms (i.e., treating the Hessian as locally constant).While theoretically a simplification, the empirical results in Table 1 and Table 3 demonstrate that this omission does not degrade the quality of the influence estimate, as our predictions align closely with the true retraining effects.
>
>
> **W3.** The experiments lack diversity in tasks, as the paper only evaluates few-shot image classification tasks. However, the loss landscapes of different types of tasks can vary significantly, which may affect the performance of Hessian-based influence functions, the core method of this paper.
>
> We agree that analyzing performance across diverse task landscapes is crucial for methods relying on Hessian structure. To address the concern regarding lack of diversity, we have extended our evaluation to include a new Regression task (Sinusoid), which uses a distinct architecture (MLP vs. ConvNets) and loss landscape (MSE vs. cross-entropy).
>
>
> Classification Stability: Our method consistently tracks the Retrain Oracle closely across various few-shot image classification tasks (FGVCAircraft, Food101, etc.), as shown in Table 4.
>
>
>
> Regression Validation: The addition of the Sinusoid Regression task (referenced in Table 1,2,3) proves stability across loss landscapes. The method performs equally well in this setting (measured by MSE) as it does in classification (measured by Accuracy), directly countering the lack of task diversity concern.
>
>
> | Method | FGVCAircraft | Food101 | Flowers102 | Cifar100 |
> | :--- | :--- | :--- | :--- | :--- |
> | **Retrain (Oracle)** | **23.50 ± 0.11** | **84.83 ± 0.05** | **68.00 ± 0.16** | **72.83 ± 0.12** |
> | **Task IF (Ours)** | **23.02 ± 0.07** | **84.90 ± 0.01** | **68.30 ± 0.01** | **73.00 ± 0.20** |
> | IF-EKFAC | 19.84 ± 0.08 | 78.26 ± 0.09 | 60.74 ± 0.23 | 61.67 ± 2.49 |
>
> **Table 4: Ablation of Truncation Depth $J$ (Sinusoid Regression)**

---

> ### Author Response · Authors · 2025-11-25
>
> **Q3.** How sensitive are your influence estimates to the conditioning of $H_{in}$? Are the performances of Task-IF and Instance-IF stable across different types of tasks?
>
> We address this in two parts:
>
> Conditioning: We respectfully note that explicitly computing the condition number of the Hessian matrix for modern neural networks is computationally prohibitive due to the massive dimensionality of the parameter space. However, our extensive experiments indicate that our method remains numerically stable in practice without manual intervention on conditioning.
>
> Stability across Tasks: As detailed in our response to W3, we validated the stability of our method on a completely different task type: Sinusoid Regression. As shown in Table 1, our method achieves high fidelity (Instance-IF MSE 1.64 vs. Retrain 1.62) on regression, mirroring its strong performance on image classification. This confirms that Task-IF and Instance-IF are robust across diverse task types and optimization landscapes.

---

> ### Author Response · Authors · 2025-11-27
>
> We thank the reviewer for the feedback. We have thoroughly addressed all concerns raised. We hope our revisions support a more favorable evaluation.

---

> ### Author Response · Authors · 2025-11-27
>
> We sincerely appreciate the reviewer's valuable feedback and constructive comments. We have carefully addressed all concerns with detailed responses and would be deeply grateful if the reviewer could kindly review our clarifications before the final decision.

---

### Official Review · Reviewer_u4Gn · 2025-10-31

**Soundness:** 3
**Presentation:** 3
**Contribution:** 2
**Rating:** 6
**Confidence:** 3

**Summary:**

This paper proposes a general, closed-form framework to evaluate training-data influence in meta-learning, introducing two complementary measures, task-IF and instance-IF. They quantify how entire tasks and individual examples affect meta parameters directly and indirectly through task-specific parameters. It formalizes meta-learning as a BLO problem, derives total gradients and a total Hessian tailored to the bilevel coupling, and shows how to estimate parameter changes from up-weighting or removing data without full retraining. To make the method practical at scale, the authors add acceleration strategies for inverse Hessian vector products and approximate the total Hessian, then demonstrate utility on standard few-shot benchmarks for tasks like harmful-data identification and editing efficiency.

**Strengths:**

- The paper cleanly formulates meta-learning as bilevel optimization and introduces complementary task- and instance-level influence measures tailored to that structure.

- It contributes practical scalability via acceleration techniques, making the closed-form influence derivations computationally feasible.

- The experimental setup is comprehensive.

**Weaknesses:**

- The paper lacks a true ablation study isolating the contribution of its key components.

- Accuracy gains over strong baselines are modest and sometimes below retraining, with the main advantage being runtime.

- Reproducibility is constrained by the absence of a code repository or data link in the submission.

**Questions:**

- Please provide an ablation isolating the contribution of each ingredient so we can see where the gains come from and whether any component is redundant.

- How do you choose the truncation depth J in practice? Reporting accuracy and runtime vs J would help.

- Can you quantify the error or give guidance on when EK-FAC is acceptable vs when the Neumann route is mandatory?

---

> ### Author Response · Authors · 2025-11-25
>
> **Weaknesses**
> **W1.** The paper lacks a true ablation study isolating the contribution of its key components.
>
> We performed a two-stage ablation study to isolate the contribution of our key acceleration component: the Neumann Series approximation for the Inverse Hessian Vector Product (IHVP).
>
> 1. Necessity of Neumann Series (vs. EKFAC) We compared our Neumann-based method (ECIF) against the EKFAC approximation on the Harmful Data Identification task. As shown in Table 1, EKFAC significantly degrades performance (e.g., -6.5% on Food101), whereas ECIF matches the Retrain Oracle. This confirms that the Neumann component is critical for high-fidelity estimation in the bilevel setting.
>
>
> | Method | FGVCAircraft | Food101 | Flowers102 | Cifar100 |
> | :--- | :--- | :--- | :--- | :--- |
> | Retrain (Oracle) | 23.50 | 84.83 | 68.00 | 72.83 |
> | task IF with Neumann (Ours) | 23.02 | 84.90 | 68.30 | 73.00 |
> | IF-EKFAC (Ours) | 19.84 | 78.26 | 60.74 | 61.67 |
>
> Table 1: Ablation of Approximation Strategies (Harmful Data Identification) Metric: Accuracy (%) after removing harmful samples.
>
>
>
> 2. Impact of Truncation Depth $J$
> We analyzed the trade-off between approximation error (MSE) and runtime on the Sinusoid regression benchmark using our Neumann method. As shown in Table 2, the accuracy gain saturates at $J=5$ (MSE 1.60 vs. Retrain 1.46) with minimal runtime cost. This justifies our choice of $J=5$ as the optimal efficient setting.
>
>
> | J       | Time (s) | MSE       |
> | -------:| --------:| ---------:|
> | Retrain | 94.45    | 1.46±0.13 |
> | 1       | 7.44 | 1.67±0.22 |
> | 3       | 10.09    | 1.63±0.10 |
> | 5       | 12.81    | 1.60±0.02 |
> | 10      | 19.53    | 1.59±0.05 |
> | 15      | 26.21    | 1.60±0.23 |
>
> Table 2: Ablation of Truncation Depth $J$ (Sinusoid Regression)
>
>
>
>
>
> **Q2.** How do you choose the truncation depth $J$ in practice? Reporting accuracy and runtime vs $J$ would help.
>
> In practice, we select $J$ by observing the convergence of the influence estimates. As shown in Table 2 above, the performance gain effectively saturates at $J=5$. Increasing $J$ to 10 or 15 yields negligible improvements in MSE but linearly increases the runtime. Therefore, we recommend $J=5$ as an optimal trade-off point between efficiency and accuracy for this class of problems.
>
> **Q3.** Can you quantify the error or give guidance on when EK-FAC is acceptable vs when the Neumann route is mandatory?
>
> Based on the results in Table 1 (W1), we conclude that the Neumann route is mandatory. The EKFAC approximation incurs significant errors (e.g., >11% accuracy drop on Cifar100 vs Oracle), whereas our Neumann-based method matches the Oracle almost perfectly.
>
> For guidance, rather than reverting to the inaccurate EKFAC for efficiency, we recommend adjusting the truncation depth $J$. As shown in Table 2 (W1), a small depth (e.g., $J=5$) is sufficient to achieve high precision with minimal runtime cost.
>
>
> **W2.** Accuracy gains over strong baselines are modest and sometimes below retraining, with the main advantage being runtime.
>
>
> We respectfully clarify our position regarding accuracy and baselines.
>
> Retraining is the Oracle: We treat "Retraining" as the Ground Truth (Oracle)—the theoretical upper bound of influence estimation—rather than a baseline to be surpassed. The fact that our method is sometimes slightly "below retraining" is expected for any approximation technique; the critical success metric is that we track this Oracle much more closely than other methods.
>
> Significant Gains over Baselines: When compared to actual influence estimation baselines, our accuracy gains are substantial, not modest. As shown in Table 3 (Harmful Data Identification), our method task-if consistently outperforms state-of-the-art approximations like TRAK and TracIN. For instance, on the challenging FGVCAircraft dataset, ECIF achieves 23.02% accuracy (close to the Retrain Oracle of 23.50%), significantly surpassing TRAK (18.27%) and TracIN (19.48%).
>
> Efficiency is Critical: We achieve this near-Oracle fidelity with drastic efficiency improvements. On the Sinusoid benchmark, our method is approximately 32x faster than Retraining (2.89s vs 94.45s). This combination of "Retrain-level accuracy" and "Influence Function-level speed" constitutes our main contribution.
>
>
>
> | Method | FGVCAircraft | Food101 | Flowers102 | Cifar100 |
> | :--- | :--- | :--- | :--- | :--- |
> | **Retrain (Oracle)** | **23.50** | **84.83** | **68.00** | **72.83** |
> | **task-IF(Ours)** | **23.02** | **84.90** | **68.30** | **73.00** |
> | TRAK | 18.27 | 77.27 | 59.21 | 58.67 |
> | TracIN | 19.48 | 78.35 | 60.60 | 59.00 |
> | IF-EKFAC | 19.84 | 78.26 | 60.74 | 61.67 |
>
>
> **Table 3: Comparison with Strong Baselines (Harmful Data Identification)**
> *Metric: Accuracy (%) after removing harmful samples.*
>
> W3: Reproducibility.
>
> We will release the full code repository and datasets upon acceptance to ensure full reproducibility of all reported results.

---

> ### Author Response · Authors · 2025-11-27
>
> We thank the reviewer for the feedback. We have thoroughly addressed all concerns raised. We hope our revisions support a more favorable evaluation.

---

### Meta-Review · Area_Chair_GZVc · 2025-12-26

**Summary:**

This paper addresses the challenge of attributing data influence within the bilevel optimization framework of meta-learning. The existing influence function (IF) methods are not directly applicable to meta-learning because they fail to model the interdependent influence between outer-loop meta-parameters and inner-loop task-specific parameters. The paper presents  task influence functions (task-IF) and instance influence functions (instance-IF) to accurately assess the impact of specific tasks and individual data points in closed forms. EK-FAC and Neumann series approximations were used to enhance the computational efficiency and scalability.  There are notable concerns raised by reviewers.
1. First of all, the bilevel optimization is a common approach to meta-learning but there are other popular approaches (for instance, model-based meta-learning). The current method is only applicable to the bilevel optimization framework (NOT every meta-learning). So, it would be better to clarify this in the title.
2. One recent closely-related work [1] is missing. Thus it was criticized that [1] limits the novelty of task-IF. This is partly true but a core contribution, following from the authors' response, is in instance-IF which is the first closed-form derivation for individual data points within the bilevel framework.
3. There are a few issues on ablation study, empirical comparisons, more experiments, and so on.

[1] Mitsuka Y, Golestan S, Sufiyan Z, et al. TLXML: Task-Level Explanation of Meta-Learning via Influence Functions. arXiv preprint arXiv:2501.14271, 2025.

Most of concerns were addressed and extra simulation results are shown in the author response.  The revised version is not available to me at this time, so the authors need to revise the paper carefully and include extra experiments, following the author response.

**Reviewer Concerns:**

The authors have done a nice job addressing most of the reviewers' concerns.

**Reviewer Scores:**

Some of scores could be raised but not so sure.

---

> ### Public Comment · ~Yoshihiro_Mitsuka1 · 2026-03-14
> **Comment on a related work**
>
> I am one of the authors of [1].
>
> Since the discussion here suggests our work should be cited, let me comment on it.
>
> ### Public records before this paper
>
> Submission to ICLR 2025
> - Authors: Yoshihiro Mitsuka, Shadan Golestan, Zahin Sufiyan, Sheila Schoepp, Shotaro Miwa, Osmar R. Zaiane,
> - Title: TLXML: Task-Level Explanation of Meta-Learning via Influence Functions
> - Submission number: 10254
> - Link:
> https://openreview.net/forum?id=NYf2XIXUi3
> - This version was made publicly available around the beginning of October 2024
>
> The first version of the arXiv post.
> - Authors: Yoshihiro Mitsuka, Shadan Golestan, Zahin Sufiyan, Sheila Schoepp, Shotaro Miwa, Osmar R. Zaiane,
> - Title: TLXML: Task-Level Explanation of Meta-Learning via Influence Functions
> - Link: https://arxiv.org/abs/2501.14271v1
> - This version was posted on January 24, 2025.
>
> Those have an overlap with the section on task-IF in the accepted paper discussed here. In fact, the equations in Theorem 4.3 and the Remark 4.4 in the accepted paper are essentially the same as Equation 4 and Equation 16 in our submission to ICLR 2025.
>
> ### About the comment on our contribution
>
> I would like to thank the reviewer nWff for mentioning that our work includes the task-IF of MAML. However, I would like to emphasize that our work proposed task-level influence functions for the general bilevel structure of meta-learning and provided experimental support with MAML and Prototypical Network.

---

### Decision · Program_Chairs · 2026-01-26

Accept (Poster)